# LOIRE: LifelOng learning on Incremental data via pre-trained language model gRowth Efficiently

**Xue Han** [*], **Yitong Wang** [*], **Junlan Feng** [†], **Wenchun Gao, Qian Hu & Chao Deng**
JIUTIAN Team
China Mobile Research Institute
Beijing, China
`{hanxueai, wangyitongyjy, fengjunlan, gaowenchun, huqianai, dengchao}@chinamobile.com`

## ABSTRACT

Large-scale pre-trained language models (PLMs) require significant computational resources to train from scratch on large volumes of data. But in the real world, emerging data from diverse sources may not be initially available for pre-training. Recent studies on lifelong learning have tried to solve this problem by exploring the use of model growth techniques to effectively incorporate new knowledge without the need for complete re-training. However, model growth approaches utilized have issues with growth operators that do not ensure strict function preservation or growth schedules that only include a few growth dimensions, reducing lifelong learning's effect. Furthermore, existing approaches often assume that emerging data has the same distribution as pre-training data, causing catastrophic forgetting of previously acquired knowledge. To address the aforementioned issues, we introduce LOIRE, a framework for lifelong learning that enables PLMs to effectively grow their capacity using incremental data. LOIRE employs growth operators for all feasible dimensions and a growth schedule to generate the optimal expansion sequence in the field of lifelong learning. Specifically, we present a novel plug-in layer growth operator with residual connections that skip the newly added layer during initial training while ensuring function preservation. We additionally propose an iterative distillation strategy for LOIRE that allows an intermediate model in the growth stages to switch between being a student and a teacher, reducing catastrophic forgetting during growth. Experiments show that LOIRE can reduce computational expenses by an average of 29.22% while retaining equivalent or better downstream performance.

## 1 INTRODUCTION

Large-scale pretrained language models (PLMs), such as GPT, BERT, and T5 (Devlin et al., 2018; Brown et al., 2020; Achiam et al., 2023; Raffel et al., 2020), have demonstrated impressive performance across various tasks. Large PLMs can require a long time and a lot of computation resources to train from scratch on huge amounts of data. However, when used in practice, PLM's performance may decline since new data from other sources frequently comes in that was not previously available for training. This alters the domain of data distribution (Guo & Yu, 2022). The mismatch hinders the widespread use of PLMs in real life. However, training from scratch is time and computation intensive, particularly if the PLM is large. Although researchers investigated involving lifelong learning (Mehta et al., 2023) for the PLM pre-training phase to prevent periodic re-training of PLMs from scratch for new data, which has the potential to reduce computing and time needs, there still remains an efficient problem unsolved. Recent research (Jin et al., 2021; Chen et al., 2023; Qin et al., 2022) have developed effective lifelong learning pipelines for PLMs by employing model growing approaches (Chen et al., 2021; Wang et al., 2023; Yao et al., 2023) to gradually increase model capacity, thereby enhancing PLMs' ability to quickly adapt to new knowledge. SOTA model growth

---

[*] These authors contributed to the work equally.
[†] Corresponding author.

strategies rely on sequential training with scaling parameters. During growth, each larger model inherits knowledge from the preceding smaller model by reusing its parameters to initialize.

Despite their success, the model growth strategies adopted for lifelong learning continue to face challenges. 1) Growth schedules, which specify when and where to expand the model structure, have mostly been examined for Transformer considering the number of layers and the width of the Feed-Forward Network (FFN) (Gong et al., 2019; Gu et al., 2020). However, devising an effective schedule that includes all possible growth dimensions remains a difficulty. 2) Growth operators, which refer to the steps taken during growth to inherit knowledge from the previous model, have to assure function preservation. Function preservation is a theoretically essential property for growth operators since it ensures that the initialized larger model behaves identically to the preceding smaller model (Chen et al., 2015). This property has proven beneficial for both knowledge inheritance (Chen et al., 2015; 2021; Shen et al., 2022) and training stability (Yao et al., 2023). Existing efforts that merely aim for non-strict function preservation may not inherit all relevant information, resulting in inadequate growth. This, in turn, has an impact on the lifelong learning effect because it creates a wide function gap during rapid growth, preventing further gains in training dynamics. 3) In a lifelong learning scenario, the data distribution is constantly shifting as new domains keep emerging (Ke et al., 2023). Existing model growth methods frequently assume that pretraining follows the same data distribution, resulting in catastrophic forgetting of previously learned knowledge.

We propose the LOIRE (**L**ifelong learning framew**O**rk on **I**ncremental data via PLMs g**R**owth **E**fficiently) to address these issues. Our contributions are listed as below:

(1) We present a novel plug-in layer growth operator that replicates the selected layers and inserts them between the original layer and the subsequent layer. This design is inspired by recent research findings that show similarity in pattern classes between neighboring Transformer layers (Delétang et al., 2023). We adopt the concept of residual connection by introducing a set of gates to intentionally skip the newly added layer in the initial training, thus ensuring the function-preserving property. We also show that this layer growth is theoretically function-preserving.

(2) In addition to the proposed layer operator, we provide a systematic definition for multi-dimensional operators that includes the hidden dimension, the FeedForward Network dimension, and the number of heads in the multi-head attention. Next, we build a growth schedule that generates an optimal growth sequence by combining multi-dimensional operators in the field of lifelong learning, allowing for efficient PLM training.

(3) Finally, we suggest an iterative distillation warmup strategy for LOIRE. According to a previous study, the difference in model size between a teacher and a student can affect the distillation performance (Mirzadeh et al., 2020). In our case, the difference in growth dimensions during every stage of growth results in a similar phenomenon. We suggest allowing an intermediate model created throughout the growth steps to switch between being a student and a teacher during the iterative distillation process. According to the iterative distillation warmup, LOIRE could accommodate the new data distribution without forgetting earlier distributions during model growth.

## 2 METHODOLOGY

Fig 1 is an overview of the whole paradigm. LOIRE combines multi-dimensional function-preserving growth operators with an optimized growth schedule. Each forward pass in the growth schedule yields a larger model by increasing the capacity of the previous smaller one and training on newly acquired data. To reduce the disparity between the data distribution before and after model growth, we use an iterative distillation strategy to warm up the larger model generated after each growth step.

### 2.1 PRELIMINARY

In this study, we focus on the Transformer architecture (Vaswani et al., 2017) that is prevalent in existing PLMs.

**Hidden states** $H^{l-1}$ represents the input for the Transformer layer $l$, which is a bi-dimensional tensor with $s$ and $h$ being the sequence and hidden dimension. When the $h$ changes, it affects every module of the Transformer structure. We overlook the position embedding in this work as it does

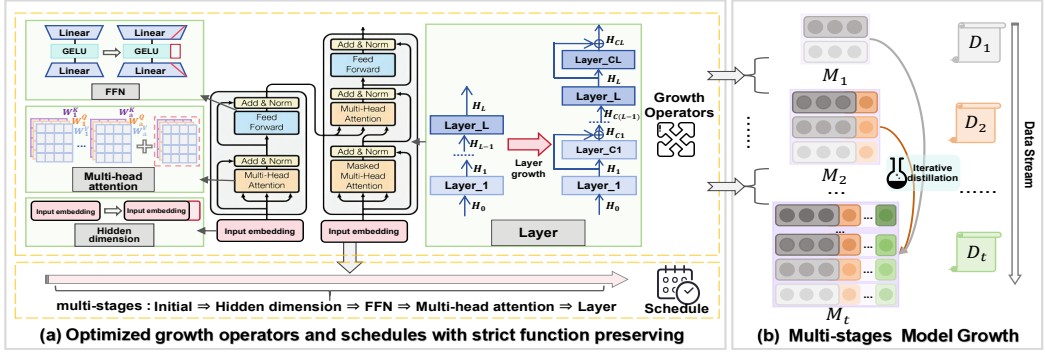

Figure 1: The overall framework of the proposed LOIRE. (a) Illustrates the growth operators and schedules over all four possible Transformer dimensions, including multi-head attention, hidden dimension, FFN, and plug-in layer growth operators. (b) Describes the multi-stage model growth procedure with an interactive distillation approach over emerging datasets.

not affect the expansion process. The hidden states are iteratively passing through the Transformer layers: $H^l_{s \times h} = Trans_l(H^{l-1}_{s \times h}), l \in [1, L]$, where L denotes the total number of the layers.

Each Transformer layer $l$ contains the modules that are important for the growth approach, which are described below:

**Multi-head attention (MHA)**: Multiple parallel self-attention heads make up MHA. The input $H$ of each layer is fed into the MHA mechanism, which can be formulated as follows:

$$K_i / Q_i / V_i = H \times W^{K/Q/V}_{head_i}$$
$$\underset{s \times d}{} \underset{s \times d}{} \underset{s \times d}{} \underset{s \times h}{} \underset{h \times d}{}$$

$$H_{head_i} = Attention(Q_i, K_i, V_i) = softmax(\frac{1}{\sqrt{d}} \times Q_i \times K_i^T) \times V_i \qquad (1)$$
$$\underset{s \times d}{}$$

$$H^{MHA}_{s \times h} = MHA(H) = [H_{head_1}, ..., H_{head_a}] \times W^O_{(a \times d) \times h}$$

where $H$ is applied to linear projection for generating queries, keys and values($Q/K/V$), utilizing different weights($W^{K/Q/V}$) for each transformation respectively. $H_{head_i}$ signifies the output of the i-th attention head with $a$ being the total number of heads. The output linear matrix $W^O$ generates the final result $H^{MHA}$, which is then delivered to the Feed-forward network.

**Feed-forward network (FFN)** is a Multi-Layer Perceptron responsible for applying a non-linear transformation to $H^{MHA}$ ($f$ is FFN's dimension of its internal representation):

$$H^{FFN}_{s \times h} = FFN(H^{MHA}) = GELU(H^{MHA}_{s \times h} \times W^{l1}_{h \times f} + b^{l1}_{s \times f}) \times W^{l2}_{f \times h} + b^{l2}_{s \times h} \qquad (2)$$

**Layer normalization (NORM)** and residual connections are interconnected, facilitating the mitigation of internal covariate shift and enhancing gradient flow in both MHA and FFN.

$$H^{NORM^{MHA}} = Layer\_norm(H + H^{MHA}) \quad H^{NORM^{FFN}} = Layer\_norm(H^{MHA} + H^{FFN}) \qquad (3)$$

where $H^{NORM^{MHA}}$ is the normalization output of $H^{MHA}$, the same for FFN module.

## 2.2 PROBLEM FORMULATION

**Lifelong learning** involves training an initial PLM $M^{(1)}$ on corpus $D_1$ and then updating it to $M^{(t)}$ with each $D_t \in \overline{D}=\{D_1, ...D_t, ....D_T\}$ to learn new knowledge while retaining old knowledge. $\overline{D} = \{D_1, ..., D_T\}$ is an incremental data stream that consists of corpus from different stages $T$. For each $D_t$, there may contain data from multiple domains. We aim to efficiently initiate $M^{(t)}$, reducing the gap with earlier $M^{(t-1)}$ in acquired knowledge.

**Lifelong learning with an efficient model growth strategy** can be further formulated as below. Growing the $M^{(t-1)}$ to $M^{(t)}$ is a process with multi-stages, which means $M^{(t-1)}$ needs go through

$K$ stages to eventually grow into $M^{(t)}$: $M_1^{(t)} \Rightarrow M_2^{(t)} ... \Rightarrow M_K^{(t)}$. Let $\mathcal{L}$ be the target loss function while $\mathcal{T}$ be the total pre-training time, the optimized objective of lifelong training can be defined as:

$$argMin_{\varepsilon}\{\mathcal{L}(\overline{\varepsilon}), \mathcal{T}(\overline{\varepsilon})\}$$

$$s.t. \quad \begin{cases} \overline{\varepsilon} = \{M_k(x;\theta_k)\}_{k=1}^K \\ \theta_k = \varphi(\theta_k^+) + \theta_{k-1}, \varphi \in \overline{\psi} \end{cases} \tag{4}$$

where $\overline{\varepsilon}$ is the multi-stage growth schedule. $\overline{\psi}$ is the growth operator set with $\psi$ as an operator for one dimension, which we will introduce in subsections 2.5 and 2.3 separately.

## 2.3 GROWTH OPERATORS WITH STRICT FUNCTION PRESERVING

We introduce the growth operator set $\overline{\psi}$ defined in equation 4. The term "growth operator" $\varphi$ denotes the actions performed during the growth phase to transfer knowledge from the prior model. The growth operator set consists of several growth operators $\varphi$, defined as $\overline{\psi} = \{\varphi | \varphi \in \psi\}$. In this work, we consider all four Transformer dimensions for growth, including layer, multi-head attention, feed-forward network, and hidden states, defined with $\psi = \{\varphi_{layer}, \varphi_{mha}, \varphi_{ffn}, \varphi_{hidden}\}$.

Function preservation is a critical property for each growth operator $\varphi$ because it ensures that the initialized larger model performs exactly the same as the preceding smaller model (Chen et al., 2015).

Specifically, suppose a PLM trained from the previous stage is represented as $M_{k-1}(x;\theta_{k-1})$ with input $x$ and parameters $\theta_{k-1}$. The growth operator $\varphi$ constrains the larger model's increased parameters $\theta_k^+$. This is achieved by choosing a new collection of parameters $\theta_k$ for the larger model $M_k$ that satisfy specific criteria defined as below:

$$\forall x, M_k(x;\theta_k) = M_k(x;\varphi(\theta_k^+) + \theta_{k-1}) = M_{k-1}(x;\theta_{k-1}) \tag{5}$$

Next, we elaborate on all four growth operators $\psi = \{\varphi_{layer}, \varphi_{mha}, \varphi_{ffn}, \varphi_{hidden}\}$ and demonstrate the validity of strict function-preserving transformations for each.

We first introduce $\varphi_{mha}, \varphi_{ffn}, \varphi_{hidden}$, leaving the layer operator $\varphi_{layer}$ introduced in the next subsection. $\varphi_{mha}, \varphi_{ffn}$, and $\varphi_{hidden}$ are defined according to Gesmundo & Maile (2023)'s work. We simplify each layer's initial input into $H$. Due to space limitations, we only present the constraints of growth operators with strict function preservation on the parameters of each module and provide proofs of function preservation in the appendix G.

**MHA growth operator** $\varphi_{mha}$ refers to the act of introducing new heads within the multi-head attention module. As mentioned in equation 1, the hyper-parameter $a$ controls the scaling of the multi-head attention dimension. When the head number increases from $a_1$ to $a_2$, we keep the weights of the former heads fixed while assigning random values to the weights of the new heads.

$$W_i^{K/Q/V} = \begin{cases} W_i^{K/Q/V} & i \le a_1 \\ any\_value & a_i < 1 \le a_2 \end{cases} \tag{6}$$

As the number of heads increases, alterations are also observed in the size of the corresponding weight matrix $W^O$ in equation 1. We set the expanded portion of $W^O$ to be a zero matrix $N$ as:

$$\underset{(a_1 \times d) \times h}{W^O} \Rightarrow \underset{(a_2 \times d) \times h}{(W^O)'} = \begin{bmatrix} \underset{(a_1 \times d) \times h}{W^O} \\ \underset{((a_2-a_1) \times d) \times h}{N} \end{bmatrix} \tag{7}$$

**FFN growth operator** $\varphi_{ffn}$ can be scaled up by increasing its internal representation's dimensionality. In equation 2, the scaling of FFN expansion is controlled by the hyper-parameter $f$. Given a Transformer layer as an example, when the FFN's hidden dimension is increasing from $f_1$ to $f_2$, we set the weights of extended $W^{l_2}$ to be $N$, while the extended part of $W^{l_1}$ and $b^{l_1}$ are initialized arbitrarily, written as $R$:

$$\underset{h \times f_1}{W^{l_1}} \Rightarrow \underset{h \times f_2}{(W^{l_1})'} = \begin{bmatrix} \underset{h \times f_1}{W^{l_1}} & \underset{h \times (f_2-f_1)}{R^{W_{l_1}}} \end{bmatrix} \quad \underset{s \times f_1}{b^{l_1}} \Rightarrow \underset{s \times f_2}{(b^{l_1})'} = \begin{bmatrix} \underset{s \times f_1}{b^{l_1}} & \underset{s \times (f_2-f_1)}{R^{W_{l_1}}} \end{bmatrix}$$

$$\underset{f_1 \times h}{W^{l_2}} \Rightarrow \underset{f_2 \times h}{(W^{l_2})'} = \begin{bmatrix} \underset{f_1 \times h}{W^{l_2}} \\ \underset{(f_2-f_1) \times h}{N} \end{bmatrix} \tag{8}$$

**Hidden dimension growth operator** $\varphi_{hidden}$ is used to expand the dimension of the representation, which is originally sent into the Transformer layers. The scaling of hidden dimension expansion is controlled by the hyper-parameter $h$. When the hidden dimension of the representation is increasing from $h_1$ to $h_2$, we set the extended portion of $H$ to be $N$:

$$\underset{s \times h_1}{H} \Rightarrow \underset{s \times h_2}{H'} = \begin{bmatrix} \underset{s \times h_1}{H} & \underset{s \times (h_2 - h_1)}{N} \end{bmatrix} \tag{9}$$

Then each module in Transformer exhibits variations in the scaling for the parameters with hidden dimension expansion.

In the MHA module, we set the extended portion of $W^O$ to be $N$, and the extended weight matrices of $K$, $Q$, and $V$ for each head are initialized randomly:

$$\underset{h_1 \times d}{W^{K/Q/V}} \Rightarrow \underset{h_2 \times d}{(W^{K/Q/V})'} = \begin{bmatrix} \underset{h_1 \times d}{W^{K/Q/V}} \\ \underset{(h_2 - h_1) \times d}{R} \end{bmatrix}$$

$$\underset{(a \times d) \times h_1}{W^O} \Rightarrow \underset{(a \times d) \times h_2}{(W^O)'} = \begin{bmatrix} \underset{(a \times d) \times h_1}{W^O} & \underset{(a \times d) \times (h_2 - h_1)}{N} \end{bmatrix} \tag{10}$$

In the FFN module, we set the extended portion of $W^{l_2}$ and $b^{l_2}$ to be $N$, while the extended $W^{l_1}$ is initialized randomly:

$$\underset{h_1 \times f}{W^{l_1}} \Rightarrow \underset{h_2 \times f}{(W^{l_1})'} = \begin{bmatrix} \underset{h_1 \times f}{W^{l_1}} \\ \underset{(h_2 - h_1) \times f}{R} \end{bmatrix} \tag{11}$$

$$\underset{f \times h_1}{W^{l_2}} \Rightarrow \underset{f \times h_2}{(W^{l_2})'} = \begin{bmatrix} \underset{f \times h_1}{W^{l_2}} & \underset{f \times (h_2 - h_1)}{N} \end{bmatrix} \quad \underset{s \times h_1}{b^{l_2}} \Rightarrow \underset{s \times h_2}{(b^{l_2})'} = \begin{bmatrix} \underset{s \times h_1}{b^{l_2}} & \underset{s \times (h_2 - h_1)}{N} \end{bmatrix}$$

According to (Gesmundo & Maile, 2023), with an expansion of the above four dimensions, the outputs of these modules remain unchangeable, given the same input. The detailed proofs are listed in Appendix G.

## 2.4 LAYER GROWTH OPERATOR

Typically, mainstream layer growth operators in the depth dimension stack the entire Transformer layer on itself until they achieve the target layer number, breaking the integrity of full-dimensional function-preserving (Gesmundo & Maile, 2023). On the other hand, recent studies have demonstrated that adjacent Transformer layers have similar pattern classes (Delétang et al., 2023), which are overlooked by existing growth operators.

Considering the factors mentioned above, LOIRE employs an optimized plug-in layer depth growth operator $\varphi_{layer}$ that replicates the selected layer and inserts it between the original layer and the subsequent layer. Motivated by the advantages of residual connections in improving the model's representation and learning capabilities, we include them in the layer growth operator to deliberately skip the newly added layer during initial training, ensuring function preservation.

For a model with $L$ layers, as illustrated in Fig.1(a), we replicate the selected layer and place it between Layer$\_l$ and Layer$\_(l + 1)$, recognizing it as Layer$\_Cl$. The output of each duplicated Layer$\_Cl$ is computed as follows:

$$H^{Cl} = \lambda_l H^l + (1 - \lambda_l) Trans_l(H^l), l \in [1, L] \tag{12}$$

In equation 12, we simulate the process of residual connections using the hyper-parameters $\overline{\lambda} = \{\lambda_1, ..., \lambda_L\}$ to assure function preservation in the layer operator. As a result, the newly added layers have the least impact on the model during the initial training phase following expansion. With $\lambda_l \in \overline{\lambda}$ setting to 1, the $l_{th}$ layer is skipped to achieve function-preserving transformation leveraging the residual connection, as proven below. Noticing that we simplify $f^{[2]}(x) = f(f(x))$ to express nested formula and $H^l$ is the input of layer $l$ defined in 2.1.

$$\begin{cases} H^{CL} = H^L \\ H^{C(L-1)} = H^{(L-1)} \\ ... \\ H^{C1} = H^1 \end{cases} \Rightarrow \begin{aligned} H^{CL} &= Trans(H^{C(L-1)}) = Trans(H^{L-1}) \\ &= Trans^{[2]}(H^{C(L-2)}) = Trans^{[2]}(H^{L-2}) \\ &... \\ &= Trans^{[L-1]}(H^{C1}) = Trans^{[L-1]}(H^1) \end{aligned} \tag{13}$$

During training, $\lambda_l$ eventually drops to 0, and residual connections vanish, resulting in the same structure as vanilla Transformers. Practically, duplicating all layers is unnecessary. You can repeat any of the layers depending on the model's growth requirements.

## 2.5 OPTIMIZED MULTI-STAGE GROWTH SCHEDULE

We introduce the details of the growth schedule $\overline{\varepsilon}$ defined in equation 4. $\overline{\varepsilon}$ consists of multiple stages of sequentially growing the model in different dimensions. As we focus on the Transformer-based architecture, dimensions include layers, MHA, FFN, and hidden states, as described in Section 2.1. Previous studies on lifelong learning have not fully explored the impact of growing sequentially throughout multiple stages(Qin et al., 2022). In order to enhance efficiency, it is advisable to create a well-optimized growth schedule that involves a sequence of gradual growth stages, each of which demonstrates rapid learning skills. This approach avoids the immediate expansion of all dimensions simultaneously.

According to Yao et al. (2023), *growing the layers and heads in later stages and having a larger hidden dimension in earlier stages can lead to better model performance*. Based on this finding, the $\overline{\varepsilon}$ is formulated as follows, where $K$ is set to 5 in this situation. We present an empirical optimization approach with ablation studies illustrated in Subsection 3.3, deferring the determination of a theoretically optimal growth schedule to future research.

$$\overline{\varepsilon} = \{M_1(x;\theta_1), M_2(x;\theta_2), M_3(x;\theta_3), M_4(x;\theta_4), M_5(x;\theta_5)\}$$
$$\theta_1 \Rightarrow \theta_2[\varphi_{hidden}(\theta_2^+) + \theta_1] \Rightarrow \theta_3[\varphi_{ffn}(\theta_3^+) + \theta_2] \Rightarrow \theta_4[\varphi_{mha}(\theta_4^+) + \theta_3] \Rightarrow \theta_5[\varphi_{layer}(\theta_5^+) + \theta_4] \tag{14}$$

## 2.6 ITERATIVE DISTILLATION WARMUP

LOIRE also designed a re-training process after each growth stage that used an iterative distillation warmup strategy to prevent catastrophic forgetting of previously learned knowledge and overfitting new data. According to a previous study, the difference in model size between a teacher and a student can affect the distillation performance (Mirzadeh et al., 2020). A similar phenomenon also holds for our model growth scenario. We propose allowing intermediary models generated throughout the growth stages to switch between student and teacher roles during the iterative distillation process. For the current interation output model $M^{(t)}$, there exists a teacher model set $M_{teacher}$ consisting of previously generated models $M_{teacher} = \{M^{(1)}, ..., M^{(t-1)}\}$. It should be noted that each $M^{(t)}$ consists of $K$ intermediate growth models. Distillation aims to minimize the weighted sum of the difference between the distributions of each model in $M_{teacher}$ and the current student model $M^{(t)}$ at the token level through Kullback-Leibler divergence. For each $M^{(j)} \in M_{teacher}(j \le t-1)$ and an input sequence $X = \{x_1, ..., x_m\}$, the knowledge distillation loss between $M^{(t)}$ and $M^{(j)}$ is calculated as:

$$L_{distill} = -\sum_i^m \sum_{v_n \in V} P(v_n | x_{i<m}, \theta_j) log \frac{P(v_n | x_{i<m}, \theta_j)}{P(v_n | x_{i<m}, \theta_t)}, j \le t-1 \tag{15}$$

where $x_{i<m}$ is from the ground truth sequence. $V$ denotes the vocabulary set and $v_n$ is the $n\_th$ word in $V$. $\theta_j$ and $\theta_t$ are parameters of the teacher $M^{(j)} \in M_{teacher}$ and student model $M^{(t)}$ respectively.

The final loss function $L_{final}$ is a combination of normal language modeling and distillation loss: $L_{final} = L_{LM} + \sum_{j \le t-1} \beta_j L_{distill\_j}$, where $\beta_j \in \overline{\beta}$ and $\overline{\beta} = \{\beta_1, ..., \beta_{t-1}\}$ is a set of hyperparameters to control the contribution of each teacher model to the final loss. After the warmup, we obtain the $M^{(t)+}$, which successfully inherits the knowledge from $\overline{D} = \{D_1, ..., D_t\}$ and is also well-prepared to be the teacher for the next model growth stage.

## 3 EXPERIMENTS

### 3.1 EXPERIMENTAL SETUP

**Pre-training Dataset.** We use five different domain datasets for growth pre-training, including the combination of Wikipedia & Book Corpus (WB)(Zhu et al., 2015), Realnews(NEWS)(Zellers et al., 2019), Amazon Reviews (REV)(He & McAuley, 2016), Biomedical papers (BIO)(Lo et al., 2019),

and Computer science papers (CS)(Lo et al., 2019), publicly available on HuggingFace Hub[1]. In each domain, we sample out 10 GB of data and divide it into pre-training and recovery memory data for distillation in a 9:1 ratio. Fig. 2 shows the correlations among these five datasets. We also use Redpajama(Weber et al., 2025) to generate initial version models for further evaluation, and then use the five domain datasets to proceed with training during lifelong learning.

**Baseline models**. We compare different kinds of baselines for both of the commonly utilized GPT-style (decoder-only) and BERT-style (encoder-decoder) PLM architectures. 1)GPT-style baselines include **GPT_S** and **GPT_L** that we train from scratch using the same 5 domain pretraining dataset mentioned before with different parameters (small: 27.59M, large: 104.78M). **GPT_R** shares settings with **GPT_L** incorporating a recovering period via retraining on reserved data for each model growth stage. We compare with other lifelong learning methods **ELLE** (Qin et al., 2022), distillation methods **Token-KD**(Cappellazzo et al., 2023) and **ER** (Chaudhry et al., 2019), which are also GPT-based. 2)BERT-style baselines include model growth methods **LiGO**(Wang et al., 2023).

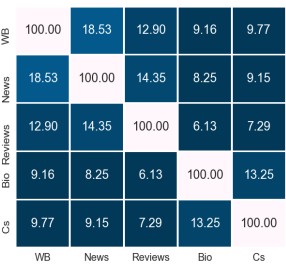

Figure 2: Vocabulary overlap (%) between five domains. Vocabularies for each domain are created by sampling the top 10K most frequent words from each domain.

**Variations of LOIRE**. There are three variations of LOIRE. LOIRE-GPT1 and LOIRE-GPT2 are both GPT-based models. We multi-stage train **LOIRE-GPT1** using our suggested model operators and schedules on five domains of pre-training data. The difference between LOIRE-GPT1 and **LOIRE-GPT2** is that for the initial stage, $M^{(1)}$ of LOIRE-GPT1 is trained from scratch. $M^{(1)}$ of LOIRE-GPT2 is initialized from a PLM starting point, which we train using Redpajama(Weber et al., 2025) following the data mixing strategy of LLAMA2(Touvron et al., 2023b). **LOIRE-Bert** shares the same initialization strategy as LOIRE-GPT2, just with a different architecture. The English wiki dataset is then utilized for growth training to ensure compliance with LIGO.

More detailed settings for both the baselines and LOIRE variations are provided in Appendix D.

**Model growth schedule setting:** Based on Subsection 2.2, we illustrate in Table 1 the growth schedule $\varepsilon$ for both GPT and BERT-based models.

Table 1: The initial PLM $M_1$ is around 27.59M parameters training from scratch. (384,1024,6,6) corresponds to the following 4 dimensions: hidden dimension, ffn dimension, head number, and layer number. Growth schedules expand one dimension at a time, as illustrated in red.

|  | $M_1$ | $M_2$ | $M_3$ | $M_4$ | $M_5$ |
|---|---|---|---|---|---|
| Model size | 27.59M | 62.25M | 71.69M | 71.69M | 104.78M |
| Growth schedule | (384,1024,6,6) | (768,1024,6,6) | (768,2048,6,6) | (768,2048,12,6) | (768,2048,12,12) |
| Growth operator | $\Rightarrow$ | $\varphi_{hidden} \Rightarrow$ | $\varphi_{ffn} \Rightarrow$ | $\varphi_{mha} \Rightarrow$ | $\varphi_{layer}$ |

## 3.2 RESULTS AND ANALYSIS

We design a set of experiments to validate LOIRE. To begin, we evaluate LOIRE's performance on the pre-training and downstream tasks. We also evaluate the function-preserving effect using the pre-training findings. Next, we compare FLOPS and wall time costs to determine training efficiency. Finally, we conducted ablation studies to investigate the separate impact of growth operators, schedules, and distillation.

**Performance on the pre-training.** We adopt **AP**(Average Perplexity) and **AP+**(Average Increased Perplexity) following (Chaudhry et al., 2019) to evaluate the pre-training performance, the lower the better. The details of metric implementations are listed in Appendix D.3.

The results are illustrated in Table 2 and Figure 3. The following insights can be drawn from Table 2: compared to all baselines, LOIRE considerably surpasses others by achieving the lowest AP+,

---

[1]https://huggingface.co/datasets

Table 2: AP represents the model's ability to learn from emerging data. AP+ represents the model's capacity to maintain old knowledge, i.e. not forget it. $M_1$-$M_5$ are models trained at multiple stages with the five-domain pre-training data. $Init$ denotes the initial ppl after growth, while $Final$ means the ppl before the next growth.

| Schedule | $M_1$ | | $M_2$ | | $M_3$ | | $M_4$ | | $M_5$ | |
|---|---|---|---|---|---|---|---|---|---|---|
| Metrics | AP | AP+ | AP | AP+ | AP | AP+ | AP | AP+ | AP | AP+ |
| *Growing $M_1$ to $M_5$ from scratch* | | | | | | | | | | |
| GPT_S | 38.69 | - | 48.83 | 23.44 | 82.15 | 96.61 | 82.30 | 117.04 | 59.59 | 83.25 |
| GPT_L | 22.43 | - | 27.69 | 12.86 | 48.87 | 55.50 | 47.75 | 64.3 | 36.01 | 48.88 |
| GPT_R | 22.43 | - | 23.78 | 3.42 | 25.92 | 6.37 | 22.61 | 9.14 | 20.66 | 8.58 |
| Token KD | 38.69 | - | 48.48 | 22.93 | 56.94 | 90.15 | 80.53 | 112.19 | 56.77 | 78.35 |
| ER | 38.69 | - | 42.28 | 9.87 | 45.30 | 14.45 | 40.24 | 21.67 | 35.94 | 19.22 |
| ELLE | 38.69 | - | 34.33 | -0.79 | 31.72 | 1.21 | 25.595 | 4.3 | 21.75 | 3.13 |
| LOIRE-GPT1 | 38.69 | - | 31.72 | **-5.32** | 29.18 | **-2.68** | 24.63 | **-0.94** | **19.19** | **-3.84** |
| *Growing $M_1$ to $M_5$ from loading an PLM* | | | | | | | | | | |
| LOIRE-GPT2 | 32.39 | - | 26.60 | -5.79 | 25.55 | -3.95 | 25.28 | -2.90 | 23.24 | -4.22 |
| *Growing $M_1$ to $M_5$ from loading an PLM* | | | | | | | | | | |
| PPL | Init | Final | Init | Final | Init | Final | Init | Final | Init | Final |
| LOIRE-Bert | 457.41 | 6.72 | 7.48 | 5.68 | 5.9 | 4.98 | 5.18 | 4.7 | 4.79 | 3.91 |

suggesting that LOIRE retains prior knowledge while acquiring new knowledge through lifelong learning. Specifically, the AP+ score decreases by 6.97 when compared to the Sota lifelong learning method, ELLE. Figure 3 provides additional evidence that LOIRE shows the most rapid decline in AP+, suggesting superior model performance under comparable training settings. It is noticed that baselines with more parameters (GPT_L and GPT_R) have a smaller AP in the early stages of training. This shows that big PLMs are extremely effective at generalization. As the training data grows, LOIRE gradually shows that it is better at keeping information from being forgotten and ultimately outperforming other baselines.

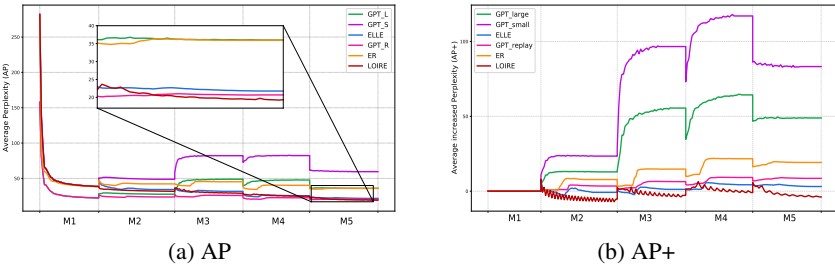

(a) AP        (b) AP+

Figure 3: The multi-stage pre-training trends curve over AP&AP+ on GPT from scratch.

LOIRE-GPT2 and LOIRE-Bert load the checkpoint trained from scratch on Redpajama to continually grow training. As shown in Table 2, experiments prove that LOIRE works both on pre-training from scratch and continually training from loading the checkpoint.

To get closer to the parameter scale of LLM models now in use in industry, particularly on the terminal side, we expand the model parameters from 177.45 M to 1.11B on the GPT structure known as LOIRE-1.1B. Specifically, we use the CC dataset[2] to train an initial 177.45 M base model for growth. Then use the same five domain datasets mentioned above to proceed with the lifelong learning. In contrast, we train a 1.11B GPT structure model (GPT-1.1B) from scratch with no growth. The other settings of GPT-1.1B are the same as LOIRE-1.1B. Table 3 shows the AP and AP+ corresponding to each expansion of LOIRE-1.1B. In the last round of expansion, the AP of LOIRE-1.1B decrease by about 6.22% compared to GPT-1.1B (same configuration as $M_5$). Experimental results show that our method is still applicable for larger models and can effectively reduce catastrophic forgetting even as the model grows.

---

[2]https://commoncrawl.org/

Table 3: AP & AP+ of LOIRE-1.1B, compared to the baseline with fixed 1.1B parameters.

| | $M_1$ | | $M_2$ | | $M_3$ | | $M_4$ | | $M_5$ | |
|---|---|---|---|---|---|---|---|---|---|---|
| Size | 177.45M | | 455.52M | | 606.57M | | 606.57M | | 1.11B | |
| Schedule | (1024,3072,16,12) | | (**2048**,3072,16,12) | | (2048,**6144**,16,12) | | (2048,6144,**32**,12) | | (2048,6144,32,**24**) | |
| Metric | AP | AP+ | AP | AP+ | AP | AP+ | AP | AP+ | AP | AP+ |
| GPT-1.1B | 14.06 | - | 18.06 | 9.73 | 27.74 | 21.50 | 34.70 | 57.38 | 20.51 | 19.52 |
| LOIRE-1.1B | 16.76 | - | 21.68 | 12.93 | 33.33 | 32.62 | **30.59** | **36.97** | **14.29** | **3.49** |

**Empirical evaluation for function preservation.** We empirically evaluate the function preservation effect over ppl of LOIRE-Bert. To the best of our knowledge, other works only prove function preservation theoretically without experiments. For LOIRE-Bert, which is growth trained using the English wiki data for all the expansion stages, we listed the ppl trends at the bottom of Table 2. The $M_1$ to $M_2$ stage implements the growth operator $\varphi_{hidden}$, resulting in a 0.76 ppl difference between the initial $M_2$ and final $M_1$. Similarly, our proposed novel growth operator $\varphi_{layer}$ has the best function preservation with only a 0.09 ppl difference between the initial $M_5$ and final $M_4$. This observation reveals that, while all of the growth operators have been theoretically proven to be stringently function-preserving, there are still deviations in the outcomes of the experiment.

Table 4: For the GPT style models, we compare the ratio of the computational costs (FLOPs) of LOIRE-GPT1 to GPT_L and GPT_R on the 5-domain dataset's pre-training. For the BERT structure, we compare the train wall time with LIGO.

| | GPT base | | | | | BERT base | |
|---|---|---|---|---|---|---|---|
| | FLOPs(%) | | | | | **Method** | **Wall Time** |
| Schedule | $M_1$ | $M_2$ | $M_3$ | $M_4$ | AVG | | |
| $\frac{LOIRE}{GPT\_L}$ | 21.20 | 78.42 | 85.69 | 85.69 | 76.20 | **LIGO** | 48h,25min |
| $\frac{LOIRE}{GPT\_R}$ | 19.08 | 70.58 | 77.12 | 77.12 | 70.78 | **LOIRE-Bert** | 28h,48min |

**Training efficiency.** From a computational cost viewpoint on GPT structure, as seen in Table 4, LOIRE achieves an average decrease of 29.22% compared to GPT_R. This leads to substantial savings in computational resources and greatly enhances efficiency. For the BERT structure, we list the train wall times of LIGO and LOIRE. LOIRE saves around 40% wall time during training. The results demonstrate that LOIRE can effectively save training time and improve training efficiency.

Table 5: The final PLMs performance after multistage model growth pre-training on representative downstream tasks including MNLI & QNLI, Hyperpartisan& Ag_news, HELPNESS & IMDB, CHEMPROT & RCT, ACL-ARC & SCIERC for domains WB, NEWS, REV, BIO, and CS respectivelyGururangan et al. (2020).

| Domain | WB | | NEWS | | REV | | BIO | | CS | | Avg |
|---|---|---|---|---|---|---|---|---|---|---|---|
| Task | MNLI | QNLI | Hyper | Ag news | HELPNESS | IMDB | CHEM | RCT | ACL-ARC | SCIERC | |
| GPT_S | 73.39 | 80.79 | 77.08 | 92.51 | 86.16 | 91.58 | 78.01 | 86.99 | 69.53 | 80.00 | 81.61 |
| GPT_L | 78.69 | 81.23 | 75.76 | 93.02 | 86.41 | 92.09 | 79.74 | 87.23 | 70.31 | **82.91** | 82.74 |
| GPT_R | 79.53 | 82.35 | 76.38 | **93.33** | 86.93 | 93.08 | 80.57 | 87.36 | 69.53 | 82.70 | 83.18 |
| Token KD | 75.58 | 80.61 | 75.16 | 92.67 | 86.32 | 91.55 | 76.12 | 86.91 | 69.53 | 78.54 | 81.30 |
| ER | 77.05 | 80.94 | 78.24 | 92.61 | **87.57** | 91.52 | 77.98 | 87.13 | 70.31 | 82.70 | 82.61 |
| ELLE | 78.12 | 83.77 | 78.75 | 93.21 | 86.59 | 92.81 | 79.98 | 87.00 | 73.43 | 79.79 | 83.35 |
| **LOIRE-GPT1** | **79.60** | **84.34** | **81.68** | 93.12 | 87.16 | **93.57** | **81.27** | **87.40** | **78.13** | 82.08 | **84.84** |

**Performance on the downstream tasks.** As shown in Table 5, we present the final performance of PLMs on representative downstream tasks. It can be observed that LOIRE achieves the highest performance across almost all downstream tasks, suggesting that the knowledge acquired during model growth can be effectively utilized and leveraged for downstream tasks. To further validate the capabilities of LOIRE, we proceed with fine-tuning BERT-style models. In Table 6, we list the performance on downstream tasks of GLUE and SAQuAD according to LIGO(Wang et al., 2023). To be noticed that the LIGO's performance is from their paper. Although we recreated LIGO using the same training data as LOIRE, the results were not comparable to their original

work. Performance demonstrates that our method achieves comparable performance while saving computational costs.

Table 6: Downstream tasks performance on GLUE and SQuAD.

| | GLUE tasks | | | | | | SQuAD tasks | Avg. |
|---|---|---|---|---|---|---|---|---|
| **Method** | **SST-2** | **MNLI** | **MRPC** | **CoLA** | **QNLI** | **QQP** | **v2.0** | **GLUE** |
| | | | Acc. | | | | F1/EM | |
| **LIGO** | 88.42 | 79.29 | 84.31 | 62.09 | 88.07 | **88.81** | 71.24/67.17 | 81.83 |
| **LOIRE-Bert** | **92.07** | **82.88** | **86.7** | **81.22** | **90.03** | 76.98 | **79.08/75.61** | **86.60** |

### 3.3 ABLATION STUDIES

**Growth schedule**: To validate our method's growth schedule, we reverse the partial expansion order. This involved converting the order of the dimensions from (hidden⇒ffn⇒head⇒layer) to (layer⇒ffn⇒head⇒hidden). We refer to this growth schedule as SCHL-Reverse and compare the performance between SCHL-Reverse and LOIRE after training on five domains in sequence. As shown in Table 7, our model, outperforms the schedule of SCHL-Reverse with growth in all dimensions, supporting the optimum sequence schedule in our design.

Table 7: The pretraining and downstream performance of SCHL-Reverse after 5 domains' multi-stage growth training, compared to our method.

| | Schedule | $M_1$ | | $M_2$ | | $M_3$ | | $M_4$ | | $M_5$ | |
|---|---|---|---|---|---|---|---|---|---|---|---|
| | | AP | AP+ | AP | AP+ | AP | AP+ | AP | AP+ | AP | AP+ |
| **Pre-training** | SCHL-Reverse | 38.69 | - | 34.64 | -3.62 | 32.47 | -0.32 | 28.43 | 1.69 | 22.23 | -2.05 |
| | LOIRE-GPT1 | 38.69 | - | 31.72 | -5.32 | 29.18 | -2.68 | 24.63 | -0.94 | **19.19** | **-3.84** |
| | Metric | acc. | acc. | acc. | acc. | acc. | acc. | acc. | acc. | acc. | acc. |
| **Downstream** | Task | **MNLI** | **QNLI** | **Hyper** | **Ag news** | **HELP** | **IMDB** | **CHEM** | **RCT** | **ACL-ARC** | **SCIERC** |
| | SCHL-Reverse | 76.85 | 82.76 | **82.06** | 92.25 | 86.89 | 91.27 | **82.48** | 87.36 | 77.34 | **83.12** |
| | LOIRE-GPT1 | **79.60** | **84.34** | 81.68 | **93.12** | **87.16** | **93.57** | 81.27 | **87.40** | **78.13** | 82.08 |

**Growth operators**: We present the impact of growth operators, replacing MHA,FFN,and hidden state growth operators by the below two methods: Random and Zero. Random refers to randomly initializing the extended portion of the parameters while keeping the remaining settings the same. Zero is identical to Random, except that it uses zero initialization rather than random initialization. From Figure 4, after initial loading, we can see that LOIRE's AP and AP+ are significantly lower than those of zero and random, proving that our operators are superior to zero and random in terms of function preservation. We leave the analysis of scheduling and distillation to a separate section in AppendixF.1 & F.2.

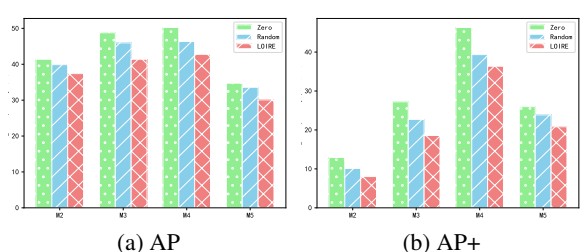

(a) AP    (b) AP+

Figure 4: AP and AP+ at the initial loading on the validation set for each seen domain after expansion.

## 4 CONCLUSION

In this study, we investigated methods of lifelong learning that incorporate current advances in the model growth field. We provide a novel plug-in layer growth operator with residual connections that skips the newly added layer during initial training, ensuring function preservation. We also introduced an iterative distillation strategy that allows a model in the growth stages to switch between being a student and a teacher, hence reducing catastrophic forgetting during dynamic learning after growth. Experiments show that LOIRE reduces computational costs by an average of 29.22% while retaining equivalent or superior downstream performance.

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

## A    LIMITATIONS

First, we wish to explore the potential for large-scale PLMs, although the largest models in our experiments have 114 million parameters because of limited computing capacity and budget, which has a gap with existing PLMs with parameters up to 65 billion, such as llama2(Touvron et al., 2023b). According to our knowledge, this constraint exists in the vast majority of research projects.

Second, although our method theoretically proves strict function preservation, there are still some deviations from the theory in the actual experimental results. Additionally, the existing research lacks empirical verification of function preserving theory, leaving us to explore and analyze this issue in the future.

Third, LOIRE performs worse on a few downstream tasks than other baselines, necessitating more analysis from the perspective of training data sample methodologies. In addition, we repeat the baselines in our experiment settings to confirm that the training procedures and data are consistent. Some of the work, such as MSG(Yao et al., 2023), has not been properly reproduced, leaving room for future research.

A further limitation is that we focus mainly in pre-training language models. Investigating techniques for other data-driven modalities, such as images and video, is also attractive.

## B    RELATED WORKS

**Efficient Pretraining** attempts to reduce FLOPs while pretraining. A few recent works focus on stagewise efficient pretraining (Panigrahi et al., 2024), progressive pretraining, or model reusing (Chen et al., 2015; 2021; Wang et al., 2023; Yao et al., 2023) by keeping the function of a pre-trained model while growing the model size. This gives a compatible larger model a starting state

with great performance. Net2Net (Chen et al., 2015) is the first to propose the concept of function-preserving transforms by expanding the width dimension by splitting neurons and growing depth by adding identity layers. bert2BERT (Chen et al., 2021) applies function-preserving concepts to the Transformer, extending Net2Net. LiGO (Wang et al., 2023) recently employs a trainable linear operator to acquire an efficient expansion strategy. Unlike other studies, our method attempts to build a full-dimension transfer that makes use of the entire smaller model.

**Lifelong learning, also known as continuing learning**(De Lange et al., 2021), has gained massive popularity in recent years. According to Gururangan et al. (2020), a second phase of in-domain pre-training (domain-adaptive pretraining) improves performance in both high and low-resource contexts. The studies conducted by Jin et al. (2021) and Wu et al. (2021) evaluate several continual learning algorithms on large PLMs using standard parameters. These studies also track the performance of these algorithms on downstream tasks. Another potential approach for continuing learning is the utilization of key-value methods derived from computer vision(Van Den Oord et al., 2017; Liu et al., 2021; Ramesh et al., 2022). The most similar work to ours is ELLE (Qin et al., 2022), which uses function-preserved model expansion and pre-trained domain prompts to efficiently pre-train for emerging data over time. The main difference is that we use a strict function preservation model growth method for all feasible domains, whereas ELLE only grows in limited domains with non-strict function preservation.

**The model growth schedule and the growth operator** are two crucial areas of research for model reuse or progressive pre-training. The growth schedule tells the model when and where to add more parameters, and the growth operator describes the steps that are taken during growth to get knowledge from the prior model. Gong et al. (2019) proposes growth schedules that stack the Transformer layers to transfer knowledge from a shallow model to a deep model. The setups that are most similar to ours are msg(Yao et al., 2023) and the work conducted by GoogleGesmundo & Maile (2023). MSG incorporates growth schedules that encompass all possible dimensions and growth operators while strictly preserving their functional integrity. Google proposes six composable transformations for gradually increasing the scale of Transformer-based neural networks while preserving functionality. However, MSG does not provide a uniform framework for the optimal integration of operators and schedules, and Google's work is lacking in schedule studies, allowing more space for our method's innovation in the field of lifelong learning.

## C  ALGORITHM

Algorithm 1 summarizes LOIRE for growing Transformer in lifelong learning.

## D  EXPERIMENTAL SETTINGS

### D.1  IMPLEMENTATION DETAILS

For GPT-based model growth, we begin by training a starting point model for 20,000 steps in the WB domain. During each model growth stage, we train the expanded model for 20,000 steps using incremental domain data. For the Bert-based structure, we train for 20,000 steps in each expansion stage, totaling 100,000 training steps on the Wiki Dataset. The initial models of both GPT and Bert are trained for 20,000 steps on the Redpajama dataset. Prior to each model growth stage, we employ a warmup strategy to retrain the model in 5,000 steps. For Bert-style structure, we train for 20,000 steps in each expansion stage, totaling 100,000 training steps on the RoBERTa structure (Liu et al., 2019) model.

For hyper-parameters, we linearly increase the $\lambda_i \in \overline{\lambda}$ value in the layer growth operator from 0 to 1 in 5000 steps per growth stage. During the iterative distillation in the warmup period, the best $\beta \in \overline{\beta}$ is set to 0.1 and vanishes after 1,000 steps. Adam is chosen as the optimizer. Our setup consists of a four-core CPU and eight NVIDIA Tesla A100 GPUs.

We list the hyper-parameters used in the GPT architecture's domain downstream experiments in Table 8. The GLUE benchmarks for downstream tasks, such as MNLI and QNLI, are based on Ott et al. (2019). SQuAD benchmarks are based on Rajpurkar et al. (2016). More downstream tasks are implemented, as detailed in Gururangan et al. (2020).

---

**Algorithm 1** LOIRE on Transformer with Growth Operators and Schedule.

---

**Input**: An initial Transformer $M^{(t-1)}$ trained on $D_{t-1}$, with Multi-head attention number being $a_1$, FFN's hidden dimension being $f_1$, Hidden dimension of input being $h_1$, and number of layers $L_1$.
**Output**: An larger Transformer $M^{(t)}$, with Multi-head attention number being $a_2$, FFN's hidden dimension being $f_2$, Hidden dimension of input being $h_2$, and number of layers $L_2$.

1: Operators: $\overline{\psi} = \{\varphi_{hidden}, \varphi_{ffn}, \varphi_{mha}, \varphi_{layer}\}$
2: Schedule: $\begin{cases} \overline{\varepsilon} = \{M_k(\theta_k)\}_{k=1}^5 \\ \theta_k = \varphi(\theta_k^+) + \theta_{k-1}, \varphi \in \overline{\psi} \end{cases}$
3: **for** $i = 1$ to $k - 1$ $(M_1^{(t)}$ to $M_5^{(t)})$ **do**
4:
5:    **if** $\overline{\psi} == \varphi_{hidden}$ **then**
6:       Hidden dimension growth: $\varphi_{hidden} : h_1 \rightarrow h_2$
7:    **end if**
8:
9:    **if** $\overline{\psi} == \varphi_{ffn}$ **then**
10:       FFN hidden dimension growth: $\varphi_{ffn} : f_1 \rightarrow f_2$
11:    **end if**
12:
13:    **if** $\overline{\psi} == \varphi_{mha}$ **then**
14:       Multi-head attention number growth: $\varphi_{mha} : a_1 \rightarrow a_2$
15:    **end if**
16:
17:    **if** $\overline{\psi} == \varphi_{layer}$ **then**
18:       Layer number growth: $\varphi_{mha} : L_1 \rightarrow L_2$
19:    **end if**
20: **end for**
21: $M^{(t-1)}(\theta_{t-1}) \Rightarrow M^{(t)}(\theta_t)$
22: Train the larger model $M^{(t)}$ on new Data $D_t$
23: Utilize the iterative distillation to generate $M^{(t)+}$ in the warmup period

---

Table 8: Hyper-parameters used in the GPT architecture's downstream experiments.

| Tasks | MNLI | Hyper | HELPNESS | CHEM | ACL-ARC |
|---|---|---|---|---|---|
| Learning Rate | $1e-5$ | $2e-5$ | $2e-5$ | $2e-5$ | $2e-5$ |
| Batch Size | 32 | 256 | 256 | 256 | 256 |
| Weight Decay | 0.1 | 0.1 | 0.1 | 0.1 | 0.1 |
| Max Epochs | 10 | 10 | 10 | 10 | 10 |
| Learning Rate Decay | Linear | Linear | Linear | Linear | Linear |
| Warmup Ratio | 0.06 | 0.06 | 0.06 | 0.06 | 0.06 |

## D.2 DETAILED SETTINGS FOR BASELINE MODELS AND LOIRE VARIATIONS

We conduct the validation to show the training acceleration on both GPT-style (decoder-only) and BERT-style (encoder-decoder) model structures. It should be noticed that all the experimental results including baselines are executed by our work.

**GPT-style baselines**:

**GPT_S** takes parameters of 27.59 million and then pretrains on each new domain data $D_t$ to learn new knowledge while maintaining the parameters invariant without a growth schedule.

**GPT_L** is similar to GPT_S but shares the same scale of parameters as our final extended model, which is 104.78 million parameters and trains from scratch over the same 5-domain pre-training dataset.

**GPT_R** is the same as GPT_L, but recovering the PLM using the subset of previously conserved corpora.

**Token-KD** is a knowledge distillation method we implemented according to (Cappellazzo et al., 2023) and continually train it while preventing forgetting from the previous model using the data from prior stages.

**ER** (Chaudhry et al., 2019) alleviates forgetting by recovering the PLM on samples from the previous training data set $D_{t-1}$ after training the PLM on the domain data $D_t$. We set the sample ratio of $D_t$ and $D_{t-1}$ to 9:1, which is consistent with our approach.

**ELLE** (Qin et al., 2022) is derived from bert2BERT, and flexibly expands an existing PLM's width and depth to improve the efficiency of knowledge acquisition using a function-preserving method, while proposing a domain prompt to stimulate the needed knowledge for downstream tasks. Different from ELLE, LOIRE employs a different function preserving method, further adopting iterative distillation to consolidate knowledge.

**LOIRE-GPT1** is trained from scratch using our suggested model operators and schedules on five domains of pre-training data.

**LOIRE-GPT2**: We initialed and pre-trained a GPT-based small PLM with 27.59M parameters from scratch using around 24GB of data sampled from Redpajama(Weber et al., 2025). The sample weights for different domains follow LLAMA2(Touvron et al., 2023b)'s data mixing strategy. LOIRE-GPT2's $M^{(1)}$ is initiated using the above-trained PLM.

**BERT-style baselines**:

**LiGO**(Wang et al., 2023) is an efficient data-driven method to map the weights between small and large models, which improves the training dynamics by learning to initialize. We re-produce this work leveraging the LIGO open-source code[3] to first train from scratch using the Redpajama dataset.

**LOIRE-Bert** shares the same training strategy as LOIRE-GPT2, just with a different architecture. LOIRE-bert is growth trained using the English Wikipedia corpus, to keep aligned with the LIGO experiment setting.

## D.3 METRICS

We utilize two metrics to evaluate how PLMs perform on the learned domains for the pre-training period, following (Chaudhry et al., 2019). **Average Perplexity (AP)** is used to measure the average perplexity of the current checkpoint $M^{(t)}$ on the validation set for each seen domain data $Domain_j(j \leq J)$. **Average Increased Perplexity (AP+)** measures the influence of current data $Domain_j$ on previous learned knowledge. Lower AP represents the model's ability to learn from emerging data. Lower AP+ represents the model's capacity to maintain old knowledge,i.e.not forget it.

For an intermediate PLM model $M^{(t)}$ during the growth stages, when learning the $j\_th$ domain, we measure the $M^{(t)}$'s perplexity $PPL_{(t),j}$ on the validation set of each seen $Domain_j$. Let $PPL_{j,j}$ be the perplexity on the $j\_th$ domain when the PLM finishes training on the $j\_th$ domains, the above metrics are calculated as follows:

$$AP = EXP(\frac{1}{J}\sum_{j=1}^{J} log PPL_{(t),j})$$
$$AP+ = \frac{1}{J-1}\sum_{j=1}^{J-1}(PPL_{(t),j} - PPL_{j,j})$$

(16)

## D.4 MODEL GROWTH SETTINGS

All of the models in our studies use vannila decode-only Transformer architectures. During model growth, we adhere to a few simple constraints contained in the existing LLM structure, as detailed in their published technical report, such as llama (Touvron et al., 2023a), qwen (Yang et al., 2024), baichuan (Yang et al., 2023), and mistral (Jiang et al., 2023). The constraints include: **1)** The hidden

---

[3]https://vita-group.github.io/LiGO/

dimension size is a multiple of 128. **2)** The hidden dimension is either 8/3 or 4 times the ffn dimension. **3)** The number of attention heads should be divisible by the hidden dimension; nevertheless, this has no effect on the model's size. With these constraints, we set the initial beginning point model with the structure (384, 1024,6 6), and the final model to be (768, 2048, 12, 12) in the main experiments.

# E  SUPPLEMENTARY EXPERIMENTS

## E.1  VISUALIZATION OF THE ATTENTION PATTERNS DURING THE MODEL GROWTH

We visualize the evolving attention patterns of a stream of interminate PLMs that are trained during the multi-stage model growth procedure. We take intermediate checkpoints after completing a phase of model growth in each domain. Next, we input the same domain data into these checkpoints to deduce the respective attention patterns. Figure 5 shows that a descendant PLM's attention patterns closely resemble those of its "ancestors," despite the model's growth and continuous training on new data. The picture's similarity demonstrates that our method successfully allows interminate PLMs to inherit and preserve the knowledge of their "ancestors."

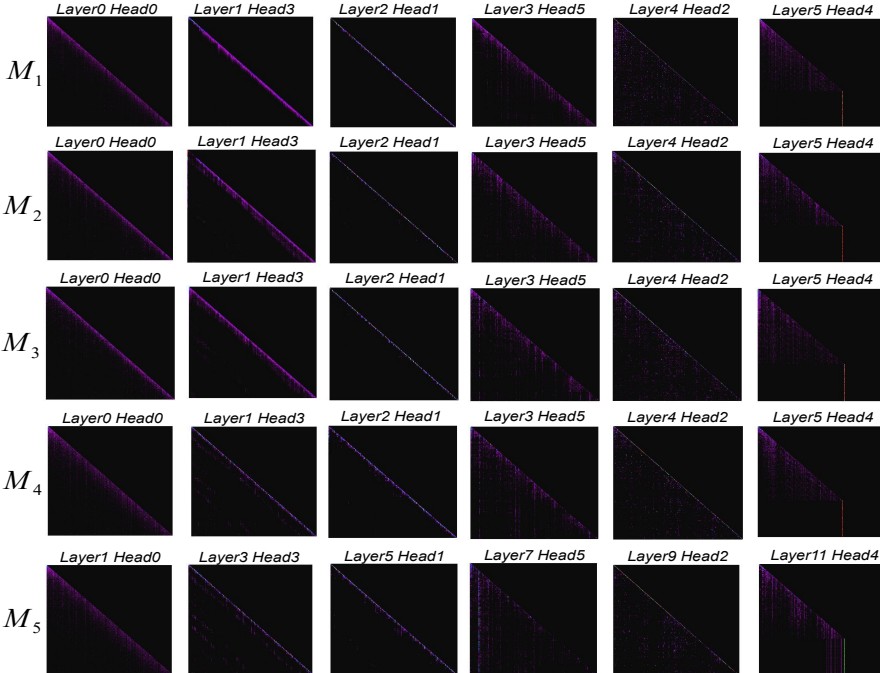

Figure 5: The visualization of the attention patterns of different attention heads in $M_1 - M_5$ after finishing training on the corresponding corpus.

## E.2  TRAINING EFFICIENCY OF LARGER PARAMETERS

We conduct experiments on the efficiency of LOIRE-1.1B mentioned in Table 3 by measuring its FLOPs and training time, and compare it with GPT-1.1B and ELLE-1.1B. As shown in the Table 9, with the increase of model parameters in lifelong stages, LOIRE-1.1B reduces FLOPs by 36.9% compared to GPT-1.1B specifically. Additionally, as illustrated in Table 3, LOIRE-1.1B does not hurt the model's performance. Therefore, with the increase of model parameters in lifelong stages, the proposed method demonstrates superiority in saving computational resources and greatly enhancing efficiency when scaling up to varying sizes.

Table 9: For the 1.1B GPT style models, we compare the FLOPs and train wall time of LOIRE-1.1B to GPT-1.1B and ELLE-1.1B on the 5-domain dataset's pre-training.

|  | $M_1$ | $M_2$ | $M_3$ | $M_4$ | $M_5$ | **Avg.** |
|---|---|---|---|---|---|---|
| **Metrics** | FLOPs(e18)/wall time(h) | | | | | |
| GPT-1.1B | (2048,6144,32,24) 34.07/23.66 | (2048,6144,32,24) 34.07/23.66 | (2048,6144,32,24) 34.07/23.66 | (2048,6144,32,24) 34.07/23.66 | (2048,6144,32,24) 34.07/23.66 | 34.07/23.66 |
| ELLE-1.1B | (1024,3072,16,12) 8.68/6.03 | (1280,3840,20,15) 13.66/9.48 | (1536,4608,24,18) 20.03/13.91 | (1792,5376,28,21) 27.93/19.39 | (2048,6144,32,24) 37.47/26.03 | 21.56/14.97 |
| LOIRE-1.1B | (1024,3072,16,12) 8.68/6.03 | (2048,3072,16,12) 20.18/14.02 | (2048,6144,16,12) 20.53/14.26 | (2048,6144,32,12) 20.53/14.26 | (2048,6144,32,24) 37.48/26.03 | **21.48/14.92** |

### E.3    INDIVIDUAL PPL OF LOIRE-GPT1 FOR EACH DOMAIN

Table 10, which lists the individual PPL of LOIRE-GPT1 for each domain as the model grows, better provides clearer insights into any knowledge degradation specific to earlier domains, and illustrates the effectiveness of our proposed lifelong method in terms of knowledge preservation.

Table 10: The individual PPL of LOIRE-GPT1 for each domain as the model grows.

|  | WB | NEWS | REV | BIO | CS |
|---|---|---|---|---|---|
| $M_1$ | 38.69 | - | - | - | - |
| $M_2$ | 33.37 | 30.16 | - | - | - |
| $M_3$ | 32.03 | 31.45 | 24.67 | - | - |
| $M_4$ | 33.01 | 30.55 | 27.13 | 13.45 | - |
| $M_5$ | 28.71 | 27.60 | 24.27 | 11.03 | 12.28 |

## F    ABLATION STUDY

The models that appeared in this section are all trained from scratch, employing GPT structure.

### F.1    LAYER OPERATOR

We propose a novel approach that increases the layer dimension by introducing a plug-in Transformer layer growth operator. This operator is designed using residual connections and is specifically optimized to maintain function-preserving quality. In order to assess its efficacy, instead of placing the duplicated layer between the original layer and the subsequent one, we replace it with a layer-stack method, which involves immediately stacking the duplicated layers on top of the existing layers, as depicted in Figure 6. Other components remain the same.

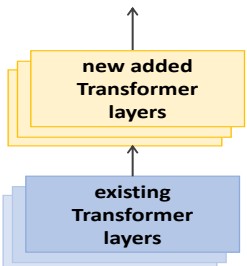

Figure 6: Layer-stack: an alternative way to replicate the existing layers.

As shown in Table 11, the AP and AP+ of our layer operator have significantly decreased, while the results of our downstream tasks are also significantly better than those of the layer-stack method. These results clearly demonstrate the ability and novelty of our proposed plug-in Transformer layer depth growth operator to preserve old knowledge.

Table 11: The pretraining performance before each expansion and the downstream performance after 5 domains' training of Layer-stack, compared to our method

| | Schedule | $M_1$ | | $M_2$ | | $M_3$ | | $M_4$ | | $M_5$ | |
|---|---|---|---|---|---|---|---|---|---|---|---|
| | | AP | AP+ | AP | AP+ | AP | AP+ | AP | AP+ | AP | AP+ |
| Pre-training | Layer-stack | 38.69 | - | 31.72 | -5.32 | 29.18 | -2.68 | 24.63 | -0.94 | 62.64 | 157.39 |
| | LOIRE-GPT1 | 38.69 | - | 31.72 | -5.32 | 29.18 | -2.68 | 24.63 | -0.94 | **19.19** | **-3.84** |
| Downstream | Metric | acc. | acc. | acc. | acc. | acc. | acc. | acc. | acc. | acc. | acc. |
| | Task | MNLI | QNLI | Hyper | Ag news | HELP | IMDB | CHEM | RCT | ACL-ARC | SCIERC |
| | Layer-stack | 77.13 | 82.48 | 79.03 | 92.87 | 86.20 | 82.31 | 81.18 | 87.36 | 75.78 | **84.79** |
| | LOIRE-GPT1 | **79.60** | **84.34** | **81.68** | **93.12** | **87.16** | **93.57** | **81.27** | **87.40** | **78.13** | 82.08 |

### F.2 EFFECT OF RECOVERING WARMUP VIA ITERATIVE DISTILLATION

In order to investigate the efficiency of the iterative distillation component, we design two comparative models. 1) Memory: takes out the knowledge distillation part of the model architecture directly and uses the subset of corpora $\overline{D}$ to get back the old knowledge. 2) Single KD: In contrast to the iterative distillation used in LOIRE, single distillation is performed using only one previous model, while the remaining components are consistent with LOIRE.

Table 12: AP and AP+ of Single KD, Memory and LOIRE with the same train wall time, we evaluate the performance after completing multi-stage training on each domain.

| Schedule | $M_1$ | | $M_2$ | | $M_3$ | | $M_4$ | | $M_5$ | |
|---|---|---|---|---|---|---|---|---|---|---|
| | AP | AP+ | AP | AP+ | AP | AP+ | AP | AP+ | AP | AP+ |
| Memory | 38.69 | - | 33.83 | -1.37 | 33.46 | 3.78 | 29.95 | 11.34 | 22.85 | 5.35 |
| Single KD | 38.69 | - | 32.81 | -2.67 | 32.50 | 3.02 | 29.41 | 11.07 | 22.19 | 4.86 |
| LOIRE-GPT1 | 38.69 | - | 31.72 | **-5.32** | 29.18 | **-2.68** | 24.63 | **-0.94** | **19.19** | **-3.84** |

As listed in Table 12, the performance of LOIRE surpasses Single KD and Memory. These results indicate that without the interactive distillation-based recovery warmup period, LOIRE's ability to retain old knowledge diminishes. Furthermore, the innovative iterative distillation method has the superior ability to incorporate previous knowledge compared to a single distillation procedure.

## G FUNCTION PRESERVING PROOFS

The following proofs are sourced from (Gesmundo & Maile, 2023). Our work differs from (Gesmundo & Maile, 2023) in that we proposes an extra-optimized layer growth operator, which we prove in Section 2. In addition, we design a growth schedule and perform experiments to verify the function preservation empirically, which (Gesmundo & Maile, 2023) only has theories.

### G.1 MHA GROWTH OPERATOR

The proof for the function-preserving transformation of head addition is as follows:

$$
\begin{aligned}
&\underbrace{[\underbrace{H_{head_1}...,H_{head_{a_2}}}_{s\times d \qquad s\times d}] \times \underbrace{(W^O)'}_{(a_2\times d)\times h}}_{s\times(a_2\times d)} \\
&= [\underbrace{\underbrace{H_{head_1}...,H_{head_{a_1}}}_{s\times d \quad s\times d},\underbrace{...,H_{head_{a_2}}}_{s\times d}}_{s\times(a_1\times d) \qquad s\times((a_2-a_1)\times d)}] \times \begin{bmatrix} \underbrace{W^O}_{(a_1\times d)\times h} \\ \underbrace{N}_{((a_2-a_1)\times d)\times h} \end{bmatrix} \\
&= [\underbrace{H_{head_1},...,H_{head_{a_1}}}_{s\times d \qquad s\times d}] \times \underbrace{W^O}_{(a_1\times d)\times h} + \underbrace{N}_{s\times h} \\
&= \underbrace{[\underbrace{H_{head_1}...,H_{head_{a_1}}}_{s\times d \qquad s\times d}]}_{s\times(a_1\times d)} \times \underbrace{W^O}_{(a_1\times d)\times h}
\end{aligned}
\tag{17}
$$

Based on the settings of the above parameters, we have:

$$\underbrace{[\underbrace{H_{head_1}}_{s \times d} ..., \underbrace{H_{head_{a_2}}}_{s \times d}]}_{s \times (a_2 \times d)} \times \underbrace{(W^O)'}_{(a_2 \times d) \times h} = \underbrace{[\underbrace{H_{head_1}}_{s \times d} ..., \underbrace{H_{head_{a_1}}}_{s \times d}]}_{s \times (a_1 \times d)} \times \underbrace{W^O}_{(a_1 \times d) \times h} \tag{18}$$

## G.2 FFN GROWTH OPERATOR

The proof for the function-preserving transformation of FFN expansion is as follows:

$$
\begin{aligned}
& GELU(\underbrace{H^{MHA}}_{s \times h} \times \underbrace{(W^{l_1})'}_{h \times f_2} + \underbrace{(b^{l_1})'}_{s \times f_2}) \times \underbrace{(W^{l_2})'}_{f_2 \times h} + \underbrace{b^{l_2}}_{s \times h} \\
&= GELU(\underbrace{H^{MHA}}_{s \times h} \times \begin{bmatrix} \underbrace{W^{l_1}}_{h \times f_1} & \underbrace{R^{W_{l_1}}}_{h \times (f_2 - f_1)} \end{bmatrix} + \begin{bmatrix} \underbrace{b^{l_1}}_{s \times f_1} & \underbrace{R^{b_{l_1}}}_{s \times (f_2 - f_1)} \end{bmatrix}) \times \begin{bmatrix} \underbrace{W^{l_2}}_{f_1 \times h} \\ \underbrace{N}_{(f_2 - f_1) \times h} \end{bmatrix} + \underbrace{b^{l_2}}_{s \times h} \\
&= GELU(\begin{bmatrix} \underbrace{H^{MHA} \times W^{l_1} + b^{l_1}}_{s \times f_1} & \underbrace{H^{MHA} \times R^{W_{l_1}} + R^{b_{l_1}}}_{s \times (f_2 - f_1)} \end{bmatrix}) \times \begin{bmatrix} \underbrace{W^{l_2}}_{f_1 \times h} \\ \underbrace{N}_{(f_2 - f_1) \times h} \end{bmatrix} + \underbrace{b^{l_2}}_{s \times h} \\
&= GELU(\underbrace{H^{MHA}}_{s \times h} \times \underbrace{W^{l_1}}_{h \times f_1} + \underbrace{b^{l_1}}_{s \times f_1}) \times \underbrace{W^{l_2}}_{f_1 \times h} + \underbrace{b^{l_2}}_{s \times h} = H^{FFN}
\end{aligned}
\tag{19}
$$

Based on the initialization of the above parameters, we have:

$$
\begin{aligned}
& GELU(\underbrace{H^{MHA}}_{s \times h} \times \underbrace{(W^{l_1})'}_{h \times f_2} + \underbrace{(b^{l_1})'}_{s \times f_2}) \times \underbrace{(W^{l_2})'}_{f_2 \times h} + \underbrace{b^{l_2}}_{s \times h} \\
&= GELU(\underbrace{H^{MHA}}_{s \times h} \times \underbrace{W^{l_1}}_{h \times f_1} + \underbrace{b^{l_1}}_{s \times f_1}) \times \underbrace{W^{l_2}}_{f_1 \times h} + \underbrace{b^{l_2}}_{s \times h} = H^{FFN}
\end{aligned}
\tag{20}
$$

## G.3 HIDDEN DIMENSION GROWTH OPERATOR

The proof for the function-preserving transformation of hidden dimension expansion for the MHA module is as follows:

$$
\begin{aligned}
& \underbrace{H'}_{s \times h_2} \times \underbrace{(W^{K/Q/V})'}_{h_2 \times d} = \begin{bmatrix} \underbrace{H}_{s \times h_1} & \underbrace{N^H}_{s \times (h_2 - h_1)} \end{bmatrix} \times \begin{bmatrix} \underbrace{W^{K/Q/V}}_{h_1 \times d} \\ \underbrace{R}_{(h_2 - h_1) \times d} \end{bmatrix} = \underbrace{H}_{s \times h_1} \times \underbrace{W^{K/Q/V}}_{h_1 \times d} \\
& \underbrace{[\underbrace{H_{head_1}}_{s \times d}, ..., \underbrace{H_{head_a}}_{s \times d}]}_{s \times (a \times d)} \times \begin{bmatrix} \underbrace{W^O}_{(a \times d) \times h_1} & \underbrace{N}_{(a \times d) \times (h_2 - h_1)} \end{bmatrix} \\
&= \begin{bmatrix} \underbrace{[\underbrace{H_{head_1}}, ..., H_{head_a}]}_{s \times (a \times d)} \times \underbrace{W^O}_{(a \times d) \times h_1} & \underbrace{N}_{s \times (h_2 - h_1)} \end{bmatrix} \\
&= \begin{bmatrix} \underbrace{H^{MHA}}_{s \times h_1} & \underbrace{N}_{s \times (h_2 - h_1)} \end{bmatrix}
\end{aligned}
\tag{21}
$$

The proof for the function-preserving transformation for the FFN module is as follows:

$$GELU((H^{MHA})'_{s \times h_2} \times (W^{l_1})'_{h_2 \times f} + b^{l_1}_{s \times f}) \times (W^{l_2})'_{f \times h_2} + (b^{l_2})'_{s \times h_2}$$

$$=GELU(\begin{bmatrix} H^{MHA}_{s \times h_1} & N_{s \times (h_2 - h_1)} \end{bmatrix} \times \begin{bmatrix} W^{l_1}_{h_1 \times f} \\ R_{(h_2 - h_1) \times f} \end{bmatrix} + b^{l_1}_{s \times f}) \times (W^{l_2})'_{f \times h_2} + (b^{l_2})'_{s \times h_2}$$

$$=GELU(\underbrace{H^{MHA}_{s \times h_1} \times W^{l_1}_{h_1 \times f} + b^{l_1}_{s \times f}}_{s \times f}) \times \begin{bmatrix} W^{l_2}_{f \times h_1} & N_{f \times (h_2 - h_1)} \end{bmatrix} + \begin{bmatrix} b^{l_2}_{s \times h_1} & N_{s \times (h_2 - h_1)} \end{bmatrix} \quad (22)$$

$$= \begin{bmatrix} \underbrace{GELU(H^{MHA}_{s \times h_1} \times W^{l_1}_{h_1 \times f} + b^{l_1}_{s \times f}) \times W^{l_2}_{f \times h_1} + b^{l_2}_{s \times h_1}}_{s \times h_1} & N_{s \times (h_2 - h_1)} \end{bmatrix}$$

$$= \begin{bmatrix} H^{FFN}_{s \times h_1} & N_{s \times (h_2 - h_1)} \end{bmatrix}$$

In the Layer normalization layer, when the hidden dimension expands, we have:

$$(H^{NORM^{MHA}})' = Layer\_norm(H'_{s \times h_2} + (H^{MHA})'_{s \times h_2})$$

$$= Layer\_norm(\begin{bmatrix} H_{s \times h_1} & N_{s \times (h_2 - h_1)} \end{bmatrix} + \begin{bmatrix} H^{MHA}_{s \times h_1} & N_{s \times (h_2 - h_1)} \end{bmatrix})$$

$$= \begin{bmatrix} Layer\_norm(H_{s \times h_1} + H^{MHA}_{s \times h_1}) & N_{s \times (h_2 - h_1)} \end{bmatrix} \quad (23)$$

$$= \begin{bmatrix} H^{NORM^{MHA}} & N_{s \times (h_2 - h_1)} \end{bmatrix}$$

Based on the settings of the above parameters, we have:

$$\begin{bmatrix} (H^{MHA/FFN}_{s \times h_2})' \end{bmatrix} = \begin{bmatrix} H^{MHA/FFN}_{s \times h_1} & N_{s \times (h_2 - h_1)} \end{bmatrix}$$

$$(H^{NORM^{MHA/FFN}}_{s \times h_2})' = \begin{bmatrix} H^{NORM^{MHA/FFN}}_{s \times h_1} & N_{s \times (h_2 - h_1)} \end{bmatrix} \quad (24)$$

The normalization of FFN layer is in a similar way. Thus, LOIRE is strictly function-preserving with LN layers.

