# OpenReview forum: "LOIRE: LifelOng learning on Incremental data via pre-trained language model gRowth Efficiently"
_ICLR.cc/2025/Conference — ICLR 2025 Poster_

### Official Review · Reviewer_MJ86 · 2024-10-31

**Soundness:** 3
**Presentation:** 3
**Contribution:** 3
**Rating:** 8
**Confidence:** 3

**Summary:**

This paper presents a method for lifelong learning of foundation models that relies on a growth strategy to allow updating of multiple dimensions of a Transformer architecture (multi-head attention, FFN, layer and hidden dimensions). Growth operators are proposed for each of these dimensions. A growth schedule, which is controlled by a hyper-parameter, is also proposed based on previous findings that "growing the layers and heads in later stages and having a larger hidden dimension in earlier stages can lead to better model performance". Experiments are run over 5 benchmark datasets, on both GPT and BERT-like architectures with increasing number of parameters. Experimental results include analyses on computation cost.

**Strengths:**

Originality: The paper is original in the sense that it proposes a new growth strategy for foundation models which consists of growing multiple dimensions of these Transformer-based models. It presents

Quality: The quality of the paper is good. There is explanation of existing concepts of new concepts, using tools such as text, diagrams and equations. There are no major points where quality needs to be

Clarity: The paper is written clearly, the concepts are presented in a reasonable flow. Experiments are also mostly clear.

Significance: There seems to be some significance of the paper from the experimental results presented against other architectures such as GPT.

**Weaknesses:**

- Significance of experiments: According to Table 1, each of the M_x steps is related to applying one of the growth operators proposed in the paper. However, the common setting in lifelong learning is grouping data(sets) into "tasks", which are learned sequentially. How is this lifelong learning idea applied in this paper?

- Significance of experiments: Similar to the previous point, in lifelong/continual learning typically measuring "forgetting" gives useful information about the "interference" experienced by the system because of learning sequentially. I do not see this reflected in this lifelong learning paper at all.

- Significance of experiments: There is no further justification as to why the particular growth schedule presented in Table 1 was selected. Are there other possible schedules? What would the results be under these schedules?

**Questions:**

- It was not entirely clear to me why there are GPT-like baselines but not BERT-like baselines to be compared against the proposed method (specifically in Table 1). Please clarify.

- In the experiments, it was not entirely clear to me how are datasets divided into n number of tasks. From Table 1, it seems that each "x" is a growth operator, so how is this related with grouping datasets into tasks? Or are all datasets used all-together from the beginning?

- Did you test all possible schedules beyond the growth schedule presented in Table 1? What are the results like?

---

> ### Author Response · Authors · 2024-11-19
>
> Thank you very much for your valuable comments and questions. I appreciate the time and effort you have put into reviewing our manuscript. Below, I address your concerns and provide further clarifications.
>
> ---
>
> **Q1**: According to Table 1, each of the M_x steps is related to applying one of the growth operators proposed in the paper. However, the common setting in lifelong learning is grouping data (sets) into "tasks", which are learned sequentially. How is this lifelong learning idea applied in this paper? how is this related with grouping datasets into tasks? Or are all datasets used all-together from the beginning?
>
> **A1**: Thank you for giving us the opportunity to elaborate. After each time we employ the operator for expansion, we utilize corresponding domain data with differing feature distributions to train the model, sequentially in the order of **WB-NEWS-REV-BIO-CS**. For example, the pretraining of $M_1$ (initial model without expansion) corresponds to WB data, and $M_5$ is trained utilizing CS data after employing the operator layer. In Table 1, by calculating the AP and AP+ of the current $M_3$, it can be validated whether $M_3$ maximizes the preservation of knowledge from the WB&NEWS domains while simultaneously learning knowledge in the REV domain.
>
> After completing the pre-training of $M_5$, to verify the model's comprehensive understanding of knowledge across the five domains, we selected two corresponding downstream experiments for each domain to further corroborate the model's performance.
>
> ---
>
> **Q2**: Significance of experiments: in lifelong/continual learning typically measuring "forgetting" gives useful information about the "interference" experienced by the system because of learning sequentially.
>
> **A2**: Thank you for your suggestions, which have contributed to the further refinement of our paper. Following the work of `[1]`, we have adopted the Forgetting Measure to assess the degree of forgetting past knowledge in the current model. We compute the forgetting measure after completing the k-th stage of pretraining. The measurement is calculated by $F_k = \frac{1}{k-1} \sum_j^{k-1} f_j^k$, where $f_k=\mathop{max}\limits_{l \in \\{1,...,k-1\\}}\ a_{l,j}-a_{k,j}$, which represents the largest gap between the past and the current accuracy for the previous downstream tasks. Table below presents the forgetting results of LOIRE-GPT1 and LORIE-nodistill (which has the same settings as LOIRE-GPT1 but without the iterative distillation warmup). We observe no significant variation in F, indicating that our proposed function-preserving operators significantly prevent forgetting, and the iterative distillation warmup further improves knowledge retention.We will include this table in subsequent versions to further refine our work.
>
> ||$M_2$|$M_3$|$M_4$|$M_5$|
> |--------| -----:|:----:|:----:|:----:|
> Forgetting measure(%)
> | LORIE-nodistill |**-4.78**|4.57|4.90|4.02|
> |LORIE-GPT1|-0.98|**-0.74**| **1.50**|**0.11**|
>
> > _[1] Chaudhry, Arslan, et al. "Efficient lifelong learning with a-gem." arXiv preprint arXiv:1812.00420 (2018)._

---

> > ### Author Response · Authors · 2024-11-20
> >
> > **Q3**: Significance of experiments: There is no further justification as to why the particular growth schedule presented in Table 1 was selected. Are there other possible schedules? What would the results be under these schedules?
> >
> > **A3**:  Firstly, as explained in Section 2.5 of the manuscript, we adopted a schedule based on empirical findings that suggest growing layers and heads in later stages and having a larger hidden dimension in earlier stages can lead to better model performance. We also conducted the ablation study as listed in Table 7 of Section 3.3, providing some evidence to support our viewpoint. The results of layer $\rightarrow$ ffn $\rightarrow$ head $\rightarrow$ hidden are not as effective as those obtained using the schedule we selected. Therefore, the growth schedule we adopted is as follows: hidden $\rightarrow$ ffn $\rightarrow$ mha $\rightarrow$ layer.
> >
> > Secondly, in our ongoing work, we are exploring the theoretical concept of achieving an optimal schedule by expressing it as an optimal path problem. We have left more details for future work regarding the page limit.
> >
> > ---
> >
> > **Q4**: It was not entirely clear to me why there are GPT-like baselines but not BERT-like baselines to be compared against the proposed method (specifically in Table 1). Please clarify.
> >
> > **A4**: Thanks for giving us the chance to clarify this concern. Recent model growth methods primarily aim to address the issue of high training costs in LLM, where the decoder-only (GPT-like) structure is the prevalent model structure. Therefore, we primarily conducted experiments on the GPT structure. We actually conducted experiments on the BERT structure (as listed in Section 3.2 RESULTS AND ANALYSIS of our manuscript), primarily to demonstrate the applicability of our method on other architectures different from GPT-2.
> > In the manuscript, Tables 4 and 6 show a full comparison of LOIRE-BERT and the BERT-structure model growth baseline LIGO in terms of both training efficiency and the classic downstream tasks of BERT.

---

> ### Author Response · Authors · 2024-11-24
>
> We sincerely thank you very much for these constructive comments and evaluation of our manuscript. As the discussion phase will be closed soon, we would like to kindly ask you to take a look at our responses and reevaluate our work based on our clarifications. Please let us know whether our response addresses your concerns or whether there is any further detail we can provide to help address these concerns.
>
> Thank you again for dedicating your time to reviewing our paper.

---

> ### Comment · Reviewer_MJ86 · 2024-11-26
> **Increased score**
>
> Thank you for your clarifications. Based on the clarifications to my questions, and those made to questions from other reviewers, I am happy to increase my score to 8. This is a good paper.

---

> > ### Author Response · Authors · 2024-11-27
> >
> > Thank you for your thoughtful feedback on our rebuttal. We will carefully follow your valuable advice and incorporate these additional results and discussions into the final version of the paper.
> >
> > Once again, we sincerely appreciate your constructive comments and support throughout the review process.

---

### Official Review · Reviewer_x51u · 2024-11-01

**Soundness:** 3
**Presentation:** 3
**Contribution:** 2
**Rating:** 6
**Confidence:** 3

**Summary:**

This paper proposes a framework called LOIRE to address the challenges of lifelong learning for pretrained language models (PLMs). The motivation comes from the need to improve the adaptability of PLMs to new, emerging data without requiring complete retraining. In real-world applications, data often evolves over time, with new domains and tasks emerge that were not part of the original pretraining data.  PLMs generally struggle with this scenario because they are typically trained once on a large dataset and then fine-tuned for specific tasks and this leads to two major problems: 1) Catastrophic Forgetting where the models are fine-tuned on new data, they tend to forget previously learned information and 2) Computational Inefficiency where retraining LLMs from scratch every time when new data emerges is computationally expensive and impractical. LOIRE addresses these challenges in lifelong learning by introducing growth operators, schedules, and distillation strategies.

**Strengths:**

LOIRE proposes several approaches to overcome the limitations given in the summary. These approaches are the strength of this work IMHO.

First one is layer growth operator that replicates selected layers and inserts them between existing layers with residual connections. The residual connections allow the newly added layers to be skipped during initial training, ensuring that the model's function is preserved. By ensuring that new layers do not interfere with the model's initial behavior, LOIRE enables smoother transitions when expanding model capacity. This is particularly important in lifelong learning scenarios where models must adapt to new data without forgetting previously acquired knowledge.

Second one is multi-dimensional growth operators that expands model capacity across multiple dimensions, such as hidden states, feed-forward networks, multi-head attention, and layers. By considering multiple dimensions, LOIRE can better adapt its expansion strategy to the specific needs of different tasks or datasets.

The main proposal is the growth schedule that determines when and where to expand the model structure across multiple stages. The schedule is designed to optimize the sequence of growth operations, ensuring that the model grows efficiently while minimizing computational costs. The results shows the significant computational savings -- an average reduction of 29.22% in expenses while maintaining or improving task performance compared to baseline methods.

And finally the iterative distillation warmup to mitigate catastrophic forgetting during model growth where intermediate models generated during growth stages switch between being students and teachers. The iterative distillation warmup strategy helps retain knowledge from previous stages while adapting to new data, ensuring that the model remains effective across both old and new tasks. This technique enhances LOIRE's ability to handle incremental data without sacrificing performance on previously learned tasks.

**Weaknesses:**

- Although these approaches are individually strong, all these process is pretty complex and i have doubts how practical is the proposed method overall in real world applications. Implementing multi-dimensional growth operators and optimizing growth schedules requires careful tuning and may be challenging for practitioners who are not familiar with these techniques. The added complexity could limit the accessibility of LOIRE for users who need simpler or more straightforward solutions for lifelong learning. It would greatly help if the authors can address these concerns.

- My main concern is the scalability of the proposed method. While LOIRE improves computational efficiency by reducing expenses by an average of 29.22%, scaling up this approach to very large models or datasets might still pose challenges. Lifelong learning frameworks need to be scalable to handle ever-growing datasets and increasingly large models. LOIRE's reliance on iterative distillation and multi-stage growth could still become computationally expensive as models grow larger. Further research may be needed to ensure that LOIRE can scale effectively without resulting in excessive costs. If the authors can provide computational complexity analyses for larger models, or conduct experiments with models of varying sizes to show how performance and efficiency scale, that would help to address this concern.

- I also have concerns about the increasing model parameters. Table-1 shows that model parameters increases significantly for each task --> M1:27.59M, M2:62.25M, M3: 71.69M, M4: 71:69M, M5: 104.78M. In reality we may observe hundreds or thousands of new tasks and if we want our model to perform well on all these how big the model will end up? That seems like a bottleneck for scalability of the approach. Can you explain what is the behaviour of parameter growth?

**Questions:**

As I mentioned in weaknesses, I have concerns about the scalability of the approach. Can you please explain how scalable the proposed method?

How well LOIRE handles significant shifts in data distribution? -- In real-world applications, new data often comes from domains that are very different from what was seen during pretraining or earlier stages of fine-tuning. It is unclear how well LOIRE would perform in such scenarios where there are drastic shifts in data distribution.

---

> ### Author Response · Authors · 2024-11-19
>
> We first want to thank the reviewer for their thorough review and largely positive comments. In particular, they highlight that the method is novel, intuitive, well-formulated, situated wrt related work, and has strong experimental results.
>
> In the rest of this response we will address the weaknesses and questions raised in the review.
>
> ---
>
> **Q1**: Challenging for practitioners who are not familiar with these techniques. The added complexity could limit the accessibility of LOIRE for users who need more straightforward solutions for lifelong learning.
>
> **A1**: Thank you for your valuable feedback. We believe that a few essential constraints imposed by the present LLM structure could decrease tuning space and simplify the implementation of the proposed model growth approach. The constraints include:
> 1. The hidden dimension size is a multiple of 128.
> 2. The hidden dimension is either 8/3 or 4 times the ffn dimension.
> 3. The number of attention heads should be divisible by the hidden dimension; nevertheless, this has no effect on the model’s size.
>
>  More details could refer to their published technical report, such as llama`[1]`, qwen`[2]`, baichuan`[3]`, and mistral`[4]`. Therefore, we believe that as long as the expanded model adheres to the aforementioned constraints, it can achieve relatively satisfactory results. Based on the provided information, users can use LOIRE as a tool to train their own models by simply designing the expanded dimensions and their corresponding expansion sizes. We will further elaborate on the constraints of the expansion in the subsequent version of our paper.
>
> > _[1] Touvron, Hugo, et al. "Llama: Open and efficient foundation language models." arXiv preprint arXiv:2302.13971 (2023)._
>
> > _[2] Yang, An, et al. "Qwen2 technical report." arXiv preprint arXiv:2407.10671 (2024)._
>
> > _[3] Yang, Aiyuan, et al. "Baichuan 2: Open large-scale language models." arXiv preprint arXiv:2309.10305 (2023)._
>
> > _[4] Jiang, Albert Q., et al. "Mistral 7B." arXiv preprint arXiv:2310.06825 (2023)._
>
> ---
>
> **Q2**: If the authors can provide computational complexity analyses for larger models, or conduct experiments with models of varying sizes to show how performance and efficiency scale.
>
> **A2**: Thanks for your insightful inquiry. We conducted experiments on the efficiency of LOIRE-1.1B by measuring its FLOPs and training time and compared it with the baselines (ELLE-1.1B and GPT-1.1B). As shown in the Table below, LORIE reduces the total FLOPs by **36.9%** when scaled up to 1.1B parameters. Additionally, as shown in Table 3 of the manuscript, LOIRE-1.1B does not hurt the model's performance. Therefore, the proposed method demonstrates superiority in saving computational resources when scaling up to varying sizes.
>
> |     | $M_1$  |  $M_2$   |  $M_3$   |  $M_4$   |  $M_5$   | Avg   |
> | --------:   | :----:   | :----:  | :----:  | :----:  | :----:  | :----:  |
> |Metrics | FLOPs(e18)/wall time(h) |FLOPs(e18)/wall time(h) |FLOPs(e18)/wall time(h) |FLOPs(e18)/wall time(h) |FLOPs(e18)/wall time(h) |FLOPs/wall time(h) | |
> | **ELLE-1.1B**    | (1024,3072,16,12) |  (1280,3840,20,15)   | (1536,4608,24,18) | (1792,5376,28,21)    |  (2048,6144,32,24)   | |
> | **ELLE-1.1B**    | 8.68/6.03 |  13.66/9.48    |  20.03/13.91   |  27.93/19.39   |   37.47/26.03    | 21.56/14.97|
> | **GPT-1.1B**    | (2048,6144,32,24) |  (2048,6144,32,24)  | (2048,6144,32,24) | (2048,6144,32,24)   |  (2048,6144,32,24) | |
> | **GPT-1.1B**    | 34.07/23.66 |  34.07/23.66    |  34.07/23.66   |  34.07/23.66   |   34.07/23.66   | 34.07/23.66|
> | **LOIRE-1.1B**    | (1024,3072,16,12) |  (2048,3072,16,12)  | (2048,6144,16,12) | (2048,6144,32,12)   |  (2048,6144,32,24) | |
> | **LOIRE-1.1B**    | 8.68/6.03 |  20.18/14.02    |  20.53/14.26   |  20.53/14.26   |   37.48/26.03   | **21.48/14.92**|

---

> > ### Comment · Reviewer_x51u · 2024-11-21
> >
> > I'm not sure if I understood comparing to GPT-1.1B. Do you apply LOIRE to some GPT models? How does it grow? Or how does LOIRE get more efficient compared to GPT1.1B? Because in LOIRE, the architecture grows when more tasks are seen, but not for GPT, right?

---

> > > ### Comment · Reviewer_x51u · 2024-11-21
> > >
> > > I'm asking for a more practical perspective. If you see 100 tasks, the model parameters will grow maybe not ~100 times, with the speed given in Table-1 (that shows that model parameters increases significantly for each task --> M1:27.59M, M2:62.25M, M3: 71.69M, M4: 71:69M, M5: 104.78M) but will keep growing. That's an important bottleneck of the proposed method, because the parameter growth is really significant. Can you at least give an approximation? Is it linear, or logistic, etc.?

---

> ### Author Response · Authors · 2024-11-20
>
> **Q3**: In reality we may observe hundreds or thousands of new tasks and if we want models to perform well on all these how big the model will end up? Can you explain what is the behaviour of parameter growth
>
> **A3**: This is a good point. First, determining how big the model will become during model growth is a new vital research topic in model growth, known as the "model growth scaling law" that has received little attention. To the best of our knowledge, just one simple empirical investigation has been conducted on this topic `[1]`. In our work in progress, we are investigating this topic theoretically, attempting to define it as an optimal path problem. We have left more details for future work regarding the page limitation.
>
> > _[1] Du W, Luo T, Qiu Z, et al. Stacking Your Transformers: A Closer Look at Model Growth for Efficient LLM Pre-Training[J]. arXiv preprint arXiv:2405.15319, 2024._
>
> ---
>
> **Q4**: How well LOIRE handle significant shifts in data distribution?
>
> **A4**: The suggested iterative distillation technique for LOIRE aims to address shifts in data distribution between real-world new data and previous training data. In the practical implementation of LOIRE, if the distribution of the new data is identical to the distribution of previous training data, merely model growth can solve the problem, and the distillation approach is unnecessary.  We also conducted ablation study to investigate the efficiency of our iterative distillation component, shown in Table 10 of Appendix F.2 .

---

> ### Author Response · Authors · 2024-11-22
>
> Q:I'm not sure if I understood comparing to GPT-1.1B. Do you apply LOIRE to some GPT models? How does it grow? Or how does LOIRE get more efficient compared to GPT1.1B? Because in LOIRE, the architecture grows when more tasks are seen, but not for GPT, right?
>
> A:Thank you for your quick response. We'd like to clarify the scenario considered within this work. We focus on the pre-training phase of PLMs. Traditional pre-training for PLMs (for example, a 177M parameter PLM) necessitates preparing all of the training data and training from scratch. However, when deployed in the world, PLMs must cope with new data that differs from the training corpora they were trained on. One alternative is to mix the emerging data with the initial training data and train from scratch to create a new PLM that can handle the new data.This is how we train GPT1.1B; typically, as the size of the training data increases, we also need to increase the PLM size to better study the expanded training corpus. However, training from scratch takes time and is computationally expensive. Another efficient approach is to utilize the initial version of the 177M PLM and scale it to 1.1B while only training the new data. This is how we proposed to get LORIE-1.1B. As a result, LORIE-1.1B may absorb emerging data more efficiently than GPT-1.1B with good performance, shown in Table in the A2 and Table 3 in the paper.

---

> > ### Comment · Reviewer_x51u · 2024-11-24
> >
> > Thank you for the explanation. It is clear now.

---

> ### Author Response · Authors · 2024-11-22
>
> Q:I'm asking for a more practical perspective. If you see 100 tasks, the model parameters will grow maybe not ~100 times, with the speed given in Table-1 (that shows that model parameters increases significantly for each task --> M1:27.59M, M2:62.25M, M3: 71.69M, M4: 71:69M, M5: 104.78M) but will keep growing. That's an important bottleneck of the proposed method, because the parameter growth is really significant. Can you at least give an approximation? Is it linear, or logistic, etc.?
>
> A:Model growth is independent of the amount of tasks. As stated in our earlier response, we concentrate on the pre-training phase of a PLM. When a considerable amount of newly collected domain-specific pre-training data is acquired linearly and chronologically, and the current PLM's parameter capacity is insufficient to accept the new knowledge included in the new domain data, lifelong learning based on model growth is required. We believe that if the increased size of task data is minor, alternative strategies, such as fine-tuning, could be more effective in this circumstance. Our proposed strategy could be combined with these techniques to address the PLM application in the real world.

---

> > ### Comment · Reviewer_x51u · 2024-11-24
> >
> > Even during pre-training phase, new data requires model growth, which is very significant. I still think this is a bottleneck.

---

> ### Author Response · Authors · 2024-11-24
>
> We sincerely thank you very much for these constructive comments and evaluation of our manuscript. As the discussion phase will be closed soon, we would like to kindly ask you to take a look at our responses and reevaluate our work based on our clarifications. Please let us know whether our response addresses your concerns or whether there is any further detail we can provide to help address these concerns.
>
> Thank you again for dedicating your time to reviewing our paper.

---

> > ### Comment · Reviewer_x51u · 2024-11-24
> > **acknowledgement**
> >
> > I increased my score to 6. But I still have concerns about parameter efficiency of the proposed method. Also, can you please explain some aspects in the new version of the paper as you already explained us in rebuttal. These are all confusing points.

---

> ### Author Response · Authors · 2024-11-25
>
> We truly appreciate the whole discussion and suggestions, which are valuable to improve our work. **We revised and re-uploaded the manuscript, as described in the general response.**
>
> Furthermore, we hope to provide an improved explanation of our understanding of the model's growth efficiency.
> We believe that the size of the model after growth can be estimated briefly by using the scaling law function, as described in `[1]` and `[2]`. According to `[1]`’s observation, the early-stopped test loss $L(N,D)$ varies predictably with the data size $D$ and model size $N$ according to the below function. Therefore, in our scenario, we know the data size $(D)$. This allows us to estimate a suitable model size $N$ for growth and further design our strategy. In fact, this is another intriguing topic that we aim to explore in our future work.
>
> $$ D \propto N^{\frac{\propto N}{\propto D}} \sim N^{0.74}$$
> $$ L(N,D)=[ (\frac{N_c}{N})^{\frac{\propto N}{\propto D}} + \frac{D_c}{D}]^{\propto D}$$
>
> > _[1] Scaling Laws for Neural Language Models. OpenAI 2020 https://arxiv.org/pdf/2001.08361.pdf_
>
> > _[2] Training Compute-Optimal Large Language Models, NIPS 2022, DeepMind. https://openreview.net/pdf?id=iBBcRUlOAPR_

---

### Official Review · Reviewer_Z3XF · 2024-11-02

**Soundness:** 3
**Presentation:** 3
**Contribution:** 3
**Rating:** 6
**Confidence:** 1

**Summary:**

This paper introduces LOIRE, a new framework, introduces comprehensive growth operators and a strategic schedule to expand PLMs effectively while preserving function. It also uses iterative distillation, allowing the model to alternate between teacher and student roles to reduce forgetting. LOIRE demonstrates a 29.22% reduction in computational costs with maintained or improved performance.

**Strengths:**

The proposed method seems novel and sound, and the extensive experiments and ablation studies demonstrate the efficacy of the method from multiple dimensions.

**Weaknesses:**

I don't see any major weakness of this paper, but that may be due to my minimal knowledge of this topic.

**Questions:**

NA

---

> ### Author Response · Authors · 2024-11-19
>
> Thanks a lot for reviewing our submitted manuscript. If you have further questions, we would be pleased to provide details to help address them.

---

### Official Review · Reviewer_NUWX · 2024-11-04

**Soundness:** 3
**Presentation:** 3
**Contribution:** 3
**Rating:** 8
**Confidence:** 3

**Summary:**

The paper introduces LOIRE, a lifelong learning framework for pre-trained language models that enables efficient adaptation to incremental data without full retraining. LOIRE tackles catastrophic forgetting and model growth challenges using a plug-in layer growth operator with residual connections for function preservation, a multi-dimensional growth schedule for optimal model expansion, and an iterative distillation strategy to retain prior knowledge. Tested on multiple benchmarks, LOIRE shows strong performance in reducing computational costs and maintaining task accuracy across evolving domains.

**Strengths:**

- Rigorous Experimental Validation: The paper describes extensive experiments conducted across various domains using multiple datasets. It also includes ablation studies that examine the effects of growth operators and schedules.
- Comprehensive Methodology: LOIRE uses a multi-dimensional growth schedule (covering dimensions like hidden size, FFN, MHA, and layers) combined with iterative distillation. This methodology is applied to manage catastrophic forgetting and model adaptation across evolving data.

**Weaknesses:**

- Limited Scalability Testing: The experiments are limited to relatively small models (up to 114M parameters), which may not fully demonstrate LOIRE’s scalability to larger PLMs, commonly used in industry. Further exploration with larger models could strengthen confidence in its applicability for high-parameter models.
- Although LOIRE’s function preservation claims are theoretically sound, empirical validation shows minor deviations. Further exploration into these deviations might strengthen the reliability of the proposed growth operators, especially in dynamic adaptation scenarios.

**Questions:**

- Have you experimented with different initialization methods for the extended parts of the model, such as the Hidden Dimension, FFN, or MHA, beyond the current method you’re using? If so, what impact did these alternatives have on model performance, function preservation, and adaptation to new data?
- Did you calculate perplexity (PPL) separately for each previous domain as the model grows, rather than averaging across them? Measuring PPL individually for each domain could provide clearer insights into any knowledge degradation specific to earlier domains.
- Did you measure LOIRE’s efficiency—such as FLOPs, training time, or memory usage—compared specifically to continual learning baselines like ELLE? Since efficiency is a key claim, it would be helpful to understand how LOIRE performs against ELLE and other lifelong learning frameworks on these metrics.

---

> ### Author Response · Authors · 2024-11-19
>
> Thank you for the detailed and insightful discussions on our paper. We hope the following clarifications could provide more clear support for our claims and help address your concerns.
>
> ---
>
> **Q1**: The experiments are limited to relatively small models (up to 114M parameters). Further exploration with larger models could strengthen confidence in its applicability for high-parameter models.
>
> **A1**: We actually scaled the LLM models to 1.11B on the GPT structure called LOIRE-1.1B, as shown in Table 3 of Section 3.2 of the original manuscript. We scale to the 1.1B parameter to more closely resemble the LLM models currently in use in industry, particularly on the terminal side. In contrast, we train a 1.11B GPT structure model (GPT-1.1B) from scratch without growth. The experimental results display that our method is still applicable for larger models and can effectively reduce catastrophic forgetting even as the model grows.
>
> ---
>
> **Q2**: Experiments with different initialization methods for the extended parts of the model? If so, what impact did these alternatives have on model performance, function preservation, and adaptation to new data?
>
> **A2**: We appreciate your highlighting the need for a more detailed and precise description of the growth operators. As already illustrated in Section 3.3 ABLATION STUDIES, we experimented with two additional initialization methods to validate the effectiveness of LOIRE’s growth operators: **Random** (randomly initializing the extended portion of the parameters) and **Zero** (using zero initialization rather than random initialization) . As shown in Figure 4 of the original manuscript, after initial loading, LOIRE’s AP and AP+ are significantly lower than those of zero and random. Specifically, the AP of LOIRE in M5 decreased by approximately **4.64 and 3.5** compared to Zero and Random, while the reduction for AP+ was **5.17 and 3.05**.
>
> ---
>
> **Q3**: Did you calculate PPL separately for each previous domain as the model grows, rather than averaging across them?
>
> **A3**: Thank you for your informative query. Due to page limitations, we were unable to include all the internal experimental results. The table below, which lists the separate PPL for each domain as the model grows, better illustrates the effectiveness of our proposed lifelong method in terms of knowledge preservation. We will include this table in subsequent versions to further refine our work.
>
> | Models/Domains|WB|NEWS|REV|BIO|CS|
> |--------:|-----:|:----:|:----:|:----:| :----:|
> |$M_1$ |38.69| -| - |-| -|
> |$M_2$|33.37|30.16| - |-|-|
> |$M_3$ |32.03|31.45|24.67|-|-|
> |$M_4$|33.01|30.55|27.13|13.45| -|
> |$M_5$|28.71|27.60|24.27|11.03|12.28|

---

> > ### Author Response · Authors · 2024-11-20
> >
> > **Q4**: Did you measure LOIRE’s efficiency—compared specifically to continual learning baselines like ELLE?
> >
> > **A4**:Thank you for bringing up this important point. We evaluated the efficiency of LOIRE-1.1B by assessing its FLOPs and training time and comparing it to baselines (ELLE-1.1B and GPT-1.1B). As shown in the Table below, with the increase in the number of lifelong stages, LOIRE demonstrates its superiority in substantially saving computational resources and greatly enhancing efficiency. Specifically, LOIRE-1.1B reduces FLOPs by 36.9% compared to GPT-1.1B. Also, Table 4 in Section 3.2 RESULTS AND ANALYSIS of the paper displays the outcomes of 100M-sized models using both GPT and BERT architectures. This is more proof that LOIRE can reduce training time and make training more effective.
> >
> > || $M_1$  |  $M_2$   |  $M_3$   |  $M_4$   |  $M_5$   | Avg   |
> > | --------:|:----:| :----: | :----:  | :----:  | :----:  | :----:  |
> > |Metrics | FLOPs/wall time(h) |FLOPs/wall time(h) |FLOPs/wall time(h) |FLOPs/wall time(h) |FLOPs/wall time(h) |FLOPs/wall time(h) | |
> > | **ELLE-1.1B**    | (1024,3072,16,12) |  (1280,3840,20,15)   | (1536,4608,24,18) | (1792,5376,28,21)    |  (2048,6144,32,24)   | |
> > | **ELLE-1.1B**    | 8.68e18/6.03 |  13.66e18/9.48    |  20.03e18/13.91   |  27.93e18/19.39   |   37.47e18/26.03    | 21.56e18/14.97|
> > | **GPT-1.1B**    | (2048,6144,32,24) |  (2048,6144,32,24)  | (2048,6144,32,24) | (2048,6144,32,24)   |  (2048,6144,32,24) | |
> > | **GPT-1.1B**    | 34.07e18/23.66 |  34.07e18/23.66    |  34.07e18/23.66   |  34.07e18/23.66   |   34.07e18/23.66   | 34.07e18/23.66|
> > | **LOIRE-1.1B**    | (1024,3072,16,12) |  (2048,3072,16,12)  | (2048,6144,16,12) | (2048,6144,32,12)   |  (2048,6144,32,24) | |
> > | **LOIRE-1.1B**    | 8.68e18/6.03 |  20.18e18/14.02    |  20.53e18/14.26   |  20.53e18/14.26   |   37.48e18/26.03   | **21.48e18/14.92**|
> >
> > ---
> >
> > **Q5**: Empirical validation shows minor deviations.
> >
> > **A5**: We believe that the discrepancy between the PPLs of the final and initial in Table 2 results from inherent differences in the distribution of validation data at different stages of growth, even though both the validation and training data are part of the Wiki dataset. Additionally, the random dropout process during the calculation of PPL can also influence the computed PPL to a certain extent. Consequently, these factors collectively contribute to the observed deviation in PPL results before and after the growth phase.

---

> > > ### Comment · Reviewer_NUWX · 2024-11-22
> > >
> > > Thank you for your response. As my concern has been addressed, I will update my score to 8.

---

> > > > ### Author Response · Authors · 2024-11-23
> > > >
> > > > We sincerely appreciate your review of our response and positive feedback.

---

### Author Response · Authors · 2024-11-25
**Response for general Concern**

## Revised Paper

In general, we express our gratitude to the reviewers for their invaluable feedback, and have revised and re-uploaded the paper based on the reviewers' suggestions. The main changes are noted in yellow. The updated part primarily includes:
+ Clarify that we are focusing on the pre-training phase of PLMs in the Introduction section.
+ Clarify a few essential constraints of dimensions during the model growth process in Appendix D.4.
+ Adding experiment of Training efficiency of models with 1.1B parameters in Appendix E.2.
+ Adding experiment of Individual PPL of LOIRE-GPT1 for each domain as the model grows in Appendix E.3.

---

### Meta-Review · Area_Chair_kbtc · 2024-12-17

**Metareview:**

The paper introduces LOIRE, a lifelong learning framework for pre-trained language models (PLMs) that efficiently adapts to incremental data without full retraining. LOIRE addresses the key challenges of catastrophic forgetting and computational inefficiency through a combination of novel growth operators with residual connections, a multi-dimensional growth schedule, and an iterative distillation strategy. The proposed approach enables gradual model expansion across hidden dimensions, feed-forward networks, multi-head attention, and layers while maintaining prior knowledge. Experiments on multiple benchmarks show that LOIRE achieves significant computational savings (up to 29.22%) while preserving or improving task performance.

Strengths
+ Rigorous Experimental Validation: Extensive experiments, including ablation studies, demonstrate the effectiveness of the method across various benchmarks and dimensions.
+ Comprehensive Methodology: LOIRE uses a multi-dimensional growth strategy combined with iterative distillation, which effectively mitigates catastrophic forgetting and ensures function preservation.
+ Novel Contributions: The paper introduces layer growth operators with residual connections, a strategic growth schedule, and iterative teacher-student distillation, enhancing PLM adaptability to new tasks.
+ Clear Presentation: The paper is well-written, with clear explanations, diagrams, and equations that aid understanding.
+ Efficiency Gains: The proposed method shows significant computational savings (29.22% on average) while maintaining or improving model accuracy.

Weaknesses
+ Scalability Limitations: Experiments are limited to relatively small models (up to 114M parameters), raising concerns about LOIRE's applicability to larger PLMs and datasets commonly used in real-world applications.
+ Complexity of Implementation: The multi-dimensional growth operators, schedules, and iterative distillation add considerable complexity, which may limit adoption for practitioners seeking simpler solutions.
+ Parameter Growth: The model parameters increase significantly with each new task, raising concerns about long-term scalability as the number of tasks grows.
+ Experiment Significance: The lifelong learning setup is not fully clarified, such as the sequential grouping of tasks or measurement of forgetting, which is a standard evaluation in continual learning.
+ Unexplored Alternatives: The paper lacks justification for the chosen growth schedule and does not explore alternative schedules or initialization methods for the extended model components.

Most concerns have been addressed by the authors during the rebuttal period.

**Additional Comments On Reviewer Discussion:**

The paper started as a borderline paper. After rebuttal, three out of four reviewers increased their ratings, leading to final ratings of 6, 6, 8, 8. Most concerns are addressed by the authors. One that remains is on the scalability when the number of tasks increases, raised by Reviewer x51u. While the authors clarified that the number of model parameters is the same with different numbers of tasks, I agree with Reviewer x51u that “Even during pre-training phase, new data requires model growth, which is very significant.” Hopefully the authors could provide more results on this issue in the camera ready version.

---

### Decision · Program_Chairs · 2025-01-22

Accept (Poster)